# Mitigation of offshore wind power intermittency by interconnection of production sites

Ida Marie Solbrekke[1,2], Nils Gunnar Kvamstø[1,2], and Asgeir Sorteberg[1,2,3]

[1]Geophysical Institute, University of Bergen, Allegaten 70, 5020 Bergen, Norway
[2]Bergen Offshore Wind Centre (BOW), University of Bergen, Norway
[3]Bjerknes Centre for Climate Research (BCCR), University of Bergen, Norway

**Correspondence:** Ida Marie Solbrekke (ida.solbrekke@uib.no)

**Abstract.** This study uses a unique set of hourly wind speed data observed over a period of 16 years to quantify the potential of collective offshore wind power production. We address the well-known intermittency problem of wind power for five locations along the Norwegian continental shelf. Mitigation of wind power intermittency is investigated using a hypothetical electricity grid. The degree of mitigation is examined by connecting different configurations of the sites. Along with the wind power smoothing effect, we explore the risk probability of the occurrence and duration of wind power shut-down due to too low/high winds. Typical large-scale atmospheric situations resulting in long term shut-down periods are identified. We find that both the wind power variability and the risk of not producing any wind power decrease significantly with an increasing array of connected sites. The risk of no wind power production for a given hour is reduced from 8.0-11.2 % for a single site to under 4 % for two sites. Increasing the array-size further reduces the risk, but to a lesser extend. The average atmospheric weather pattern resulting in wind speed that is too low (too high) to produce wind power is associated with a high- (low-) pressure system near the production sites.

## 1 Introduction

Renewable power generation from various sources is continuously increasing. This is a desired development due to, among others things, emission goals that are linked to mitigation of global warming. Offshore wind power, and especially floating offshore wind power, is only in an initial phase compared to other more mature and developed energy sources. A study by Bosch et al. (2018) has found the global offshore wind energy potential to be 329.6 $TWh$, with over 50 % of this potential being in deep waters ($> 60\,m$). These numbers underline the need to take advantage of the floating offshore wind energy source with a view to addressing the continuous growth in global energy consumption.

Exploiting the offshore wind energy potential introduces a number of challenges, one of which is the variable nature of the energy source. The wind varies on both spatial and temporal scales, ranging from small features existing for a few seconds to large and slowly evolving climatological patterns. This intermittency results in a considerable variability on different time and spatial scales, leading to highly fluctuating power production and even power-discontinuities of various durations.

Fluctuating wind power production is shown to be dampened by connecting dispersed wind power generation sites (Archer and Jacobson (2007); Dvorak et al. (2012); Grams et al. (2017); Kempton et al. (2010); Reichenberg et al. (2014, 2017);

St. Martin et al. (2015)). This smoothing effect was demonstrated as early as 1979 by Kahn (1979), who evaluated the reliability of geographically distributed wind generators in a California case-study. As weather patterns are heterogeneous, the idea behind connecting wind farms that are situated far apart is that the various sites will experience different weather at a certain time. There is therefore potential to reduce wind power variability as wind farms, unlike single locations, are area-aggregated.

Previous studies examine this smoothing effect almost exclusively over land (Archer and Jacobson (2007); Grams et al. (2017); Kahn (1979); Reichenberg et al. (2014, 2017); St. Martin et al. (2015)). For example, Reichenberg et al. (2014) presented a method to minimize wind power variability using sequential optimization of site location applied to the Nordic countries and Germany. They found that by using optimal aggregation the coefficient of variability CV ($CV = \frac{\sigma}{\mu}$) was reduced from 0.91 to 0.54; meaning that wind power variability was substantially reduced when utilizing the fact that the connected production sites were located far apart and in different wind regimes. In addition to intermittency reduction, combining a number of wind power sites situated throughout Europe has also resulted in a reduced number of low-power events. Reichenberg et al. (2017) focused on minimizing the variables related to wind power variability and maximizing the average wind power output and found that periods of low output were almost completely avoided.

A few studies have examined the smoothing effect of production sites distributed over the ocean (Dvorak et al. (2012); Kempton et al. (2010)). Kempton et al. (2010) studied the stabilization of the wind power output by placing the production sites in an optimal meteorological configuration and connecting them. They used data from 11 (more or less) meridionally oriented meteorological stations spanning 2,500 km along the east coast of the US. They concluded that connecting all 11 sites resulted in a slowly changing wind power output, in addition to a production that rarely reached either full or zero wind power output. Dvorak et al. (2012) also used the east-coast of the US as study area, but at shallow water depths ($\leq 50$ m), in an attempt to identify an ideal offshore wind energy grid in terms of, among other things, a smoothed wind energy output and a reduced hourly ramp rate and hours of zero power. They found that by connecting all four farms included in the study the power output was smoothed and the hourly zero-power events were reduced from 9 % to 4 %. They also found that wind power production in regions driven by both synoptic-scale storms and mesoscale sea breeze events experienced a substantial reduction in low/zero-production hours and in the amplitude of the hourly ramp rates when all four farms were connected, compared to production from single farms.

This study is based on 16 years of wind observations from a unique string of sites along the Norwegian coast. We analyze the potential intermittency reduction of wind power output over open ocean by potentially connecting up to five power producing sites in different combinations. The water depth at these locations ranges from 75 m at Ekofisk and increases to over 350 m at Norne, which means that floating offshore wind is more or less the only option. We approximate the wind power output by transforming hourly observed wind speed observations to wind power output through a conversion function. Along with the smoothing effect we also investigate statistical wind power characteristics as a function of production site combinations. Additionally, we quantify the potential reduction in the occurrence and duration of shut-down events (no production of wind power).

Identifying typical atmospheric weather conditions that often result in long-term zero wind power production is crucial. During these shut-down events the power demand has to be covered by other energy sources or through energy storage systems

such as hydrogen or batteries. It is therefore important to map and identify these critical weather patterns, particularly if wind is likely to constitute a large share of the global energy and electricity mix. Kempton et al. (2010) examined a few high- and low-power events using reanalysis data to gain an insight into the corresponding large-scale atmospheric situation. In contrast to Kempton's work we have, by examining the composition of the atmospheric situations related to zero-events, revealed the typical (composite mean) atmospheric condition related to zero-events caused by "too low" and "too high" wind speed.

The paper is structured as follows: Section 2 includes the data and the corresponding post-processing, section 3 describes the methods used, section 4 presents the results of the study and discusses the results. Finally, section 5 summarizes the main results in bullet points.

## 2  Data and post-processing

In this exploratory study we examine the effect of collective wind power at five locations (oil and gas platforms) along the coast of Southern Norway. The data sites included in this study are Ekofisk (ek), Sleipner (sl), Gullfaks C (gf), Draugen (dr), and Norne (no) (see Fig. 1 for location and Table 2 for further information on the various sites).

| Platform | Abbreviation | Lat | Lon | Sensor | Height |
|----------|--------------|-------|------|--------|--------|
| Ekofisk | ek | 56.55 | 3.21 | A | 69/103 |
| Sleipner | sl | 58.36 | 1.91 | A | 136 |
| Gullfaks C | gf | 61.22 | 2.27 | B | 141 |
| Draugen | dr | 64.35 | 7.78 | A | 78 |
| Norne | no | 68.01 | 8.07 | A | 45 |

**Table 1.** Name, abbreviation and location (in latitude and longitude) of each site, as well as the height ($m.a.s.l.$) of the chosen wind sensor (A or B), giving the representative wind speed time series for each site. The chosen sensor at Ekofisk (ek) has two heights since the sensor was moved during the period 2000-2016.

The observed data constitute a unique data set retrieved from the Norwegian Meteorological Institute, and the time series covers the 16-year period between 2000 and 2016. We use hourly 10-min average values[1]. The observed data at each site underwent a quality check, both automatic and visual. Some of the values in the data sets were recorded as NaN (not a number). In addition, some observed wind speed values were regarded as non-physical and replaced with NaN in the time series prior to the analysis. If the data point fell into one of the three categories below, it was flagged and replaced by NaN:

- Observations with a wind speed tendency $\frac{\delta u}{\delta t} \geq 15 \ ms^{-1}$ over each of two consecutive hours - a spike in the wind speed time series.

- Observations dropping to zero from $u \geq 5 \ ms^{-1}$ in one hour

[1]10-minute average: the average of 5 min before and 5 min after every hour

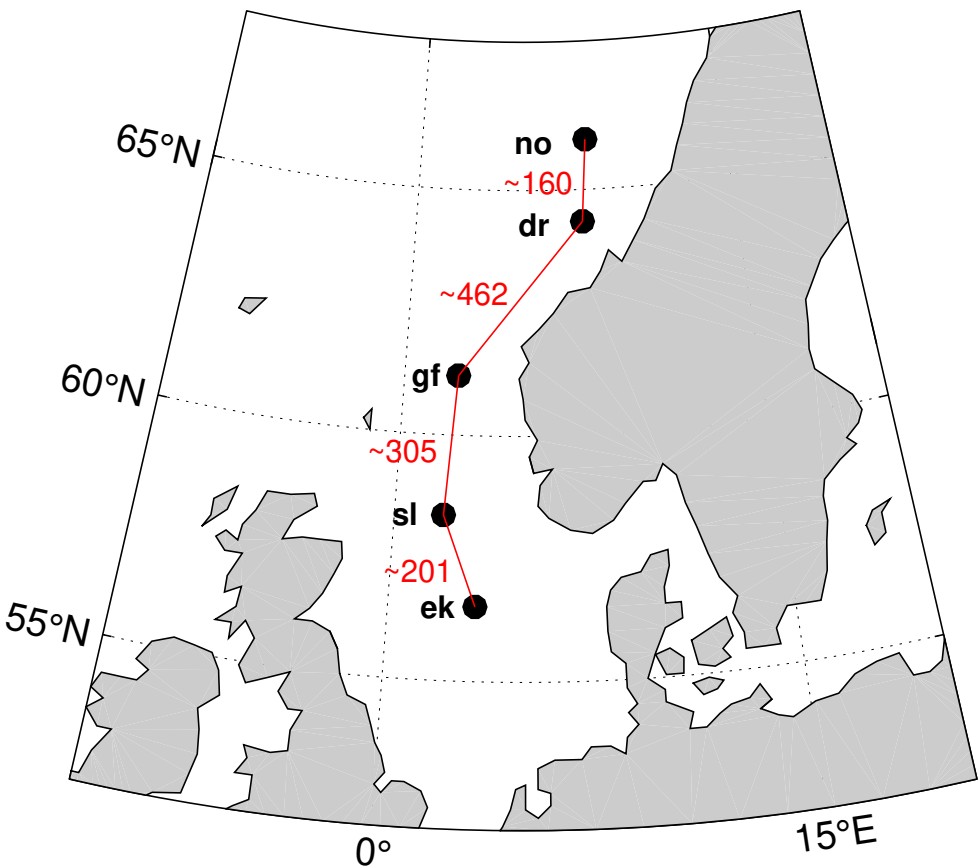

**Figure 1.** Position of the five sites and the distance (km) between them (red lines). Abbreviations: ek = Ekofisk, sl = Sleipner, gf = Gullfaks C, dr = Draugen, and no = Norne.

– Observations of $u_t = 0$ surrounded by NaN.

At each platform two anemometers were mounted at different heights, measuring the wind speed and direction. The two anemometers record two separate time series, and one of the data sets was selected to represent the wind conditions at the site in question. When choosing the most representative wind speed time series, the data set had to fulfill certain criteria, namely:

     – Contain most valid observations after the flagging-procedure above.

– Have the highest correlation with NORA10 reanalysis data, i.e. nearest grid-point with wind speed at 100 m. (Reistad et al. (2011)).

As an approximation, we can estimate the wind speed at a height level z2 by extrapolation of the wind speed from height z1 by the following relation:

$$u_{z2} = u_{z1}\left(\frac{z_2}{z_1}\right)^{\alpha}, \tag{1}$$

where $u_{z1}$ and $u_{z2}$ are the wind speed at heights z2 and z1, respectively. $\alpha$ is the power law exponent modifying the shape and steepness of the vertical wind speed profile, depending on both the surface roughness and the atmospheric stability (Emeis (2018)). Setting z2 to be the hub height at $100\ m.a.s.l$, z1 to be the height of the wind sensors and $\alpha = 0.12$ we obtain the extrapolated wind speeds at the hub height for each site. This power law is used and discussed by Borresen (1987) and Barstad et al. (2012), among others.

After replacement of the physically unrealistic observations with NaN, and the aforementioned height extrapolation and correlation check with the NORA10, we selected the wind speed data set assumed to be the most representative for the site in question. In the following sections all estimates involve only wind speed, since we assume that turbine technology allows utilization of wind power to be independent of wind direction. Moreover, we performed calculations for one turbine at each site, since we assumed that the park effect was not relevant for the results obtained.

## 3 Method

### 3.1 Estimation of wind power

The maximum part of the kinetic wind energy per time unit passing the area spanned by a wind turbine that can be utilized, is defined as the wind power, $P$, see e.g. Jaffe and Taylor (2019), and is written as:

$$P = \beta \frac{1}{2} \rho A u^3, \tag{2}$$

where $\beta$ is the Betz limit, $\rho$ is the density of the air, $A$ is the swept area of the rotors, and $u$ is the wind speed.

An analysis of the actual or theoretical wind power potential would involve an analysis of the time series of $P$. However, a more practical approach is needed as current technology only allows turbines to produce power at certain wind speed intervals. Power production starts when the wind exceeds a "cut-in" value $u_{ci}$. Subsequently, total wind power production, $P_w^T$, increases according to $P_w^T = \beta \frac{1}{2} \rho A(u^3 - u_{ci}^3)$ until u reaches the rated wind speed value $u_r$. The rated wind speed denotes the transition limit where the turbine starts to produce at rated power. For higher wind speeds, $u > u_r$, $P_w^T$ is kept constant until $u$ reaches a "cut-out" value, $u_{co}$. Thereafter the production is terminated ($P_w^T = 0$) abruptly with increasing wind. This abrupt termination of wind energy due to too high winds is a storm shutdown and is called "storm control". In practice, the described production regime for $u > u_r$ is brought about by instantaneous pitching of turbine blades. This pitching allows a portion of the energy to pass through the blades without utilization, and is done to shelter the turbines from the harsh drag force and to minimize the turbines maintenance. Consequently, the maximum power outtake for a turbine occurs when $u_r \leq u < u_{co}$, and can be written as $P_w^{max} = const = \beta \frac{1}{2} \rho A(u_r^3 - u_{ci}^3)$. In practice, $P_w^{max}$ is the installed capacity. In order for our results to be as general as possible in the rest of the paper, and since the turbine park at each site is only imaginary and of unknown capacity, we normalize

the power calculations $P_w = \frac{P_w^T}{P_w^{max}}$. The technological characteristics of the turbines thus results in a transformation $P \rightarrow P_w$, which can be written as:

$$P_w = \begin{cases} 0, & u < u_{ci} \\ \frac{u^3 - u_{ci}^3}{u_r^3 - u_{ci}^3}, & u_{ci} \leq u < u_r, \\ 1, & u_r \leq u < u_{co} \\ 0, & u \geq u_{co}. \end{cases} \tag{3}$$

where $u$ is the wind speed data, $u_{ci} = 4 \ ms^{-1}$ is the cut-in wind speed, $u_r = 13 \ ms^{-1}$ is the rated wind speed, and $u_{co} = 25$ $ms^{-1}$ is the cut-out wind speed. These numbers are retrieved from the SWT-6.0-154 turbines used in Hywind, Scotland (AG, 2011). The SWT-6.0-154 turbines were selected as they are the turbines used in the World's first commercially operating floating wind farm off the coast of Scotland.

## 4   Results and discussion

### 4.1   Wind speed and wind power characteristics

Some statistical quantities are studied to reveal both wind speed and wind power characteristics: For the wind speed the Weibull mean ($\mu$) and standard divation ($\sigma$), together with the scale and shape parameters (a and b) are calculated (see upper panel of Fig. 2 for an example wind speed distribution). After conversion of wind speed to wind power via Eq. 3 the data obtain an
entirely different distribution (see lower panel of Fig. 2 for an example wind power distribution). Calculating the arithmetic mean and standard deviation will most likely not result in values representing the typical wind power output and the associated variability. Instead, we use the median ($q_2$) and inter-quartile range (IQR) as a measure of the middle value and the spread in the data, respectively. Both $q_2$ and IQR are independent of the data distribution which makes them adequate choices to represent the statistical characteristics of the wind power data.
The mean wind speed values ($\mu$) for the five sites demonstrate that the potential for wind power harvestation is very good, with the mean ranging from $9.97 \ ms^{-1}$ to $11.25 \ ms^{-1}$. Zheng et al. (2016) regarded the wind speed at 90 $m.a.s.l.$ in the Norwegian Sea and the North Sea as "superb" and ranked it in the highest wind-category (category 7) with the potential to produce more than $400 \ Wm^{-2}$ of wind energy. By comparison, many of the wind parks already operating in the Yellow Sea (east of China) are only ranked in categories 4-6, ranging from "good" to "outstanding", with the potential to produce $200 - 400$
$Wm^{-2}$, respectively.
The As mentioned earlier, the wind power intermittency is a huge problem due to the balancing-difficulties and high economic costs related to a fluctuating power output. Among the five platforms, Ekofisk (ek) has the lowest variability, with a standard deviation of $5.3 \ ms^{-1}$. Gullfaks C (gf) is the site with the highest variability, where $\sigma = 6.0 \ ms^{-1}$. "a" is the scale parameter giving the height and width of the Weibull distribution. A large (small) scale parameter indicates a wide and low (high and

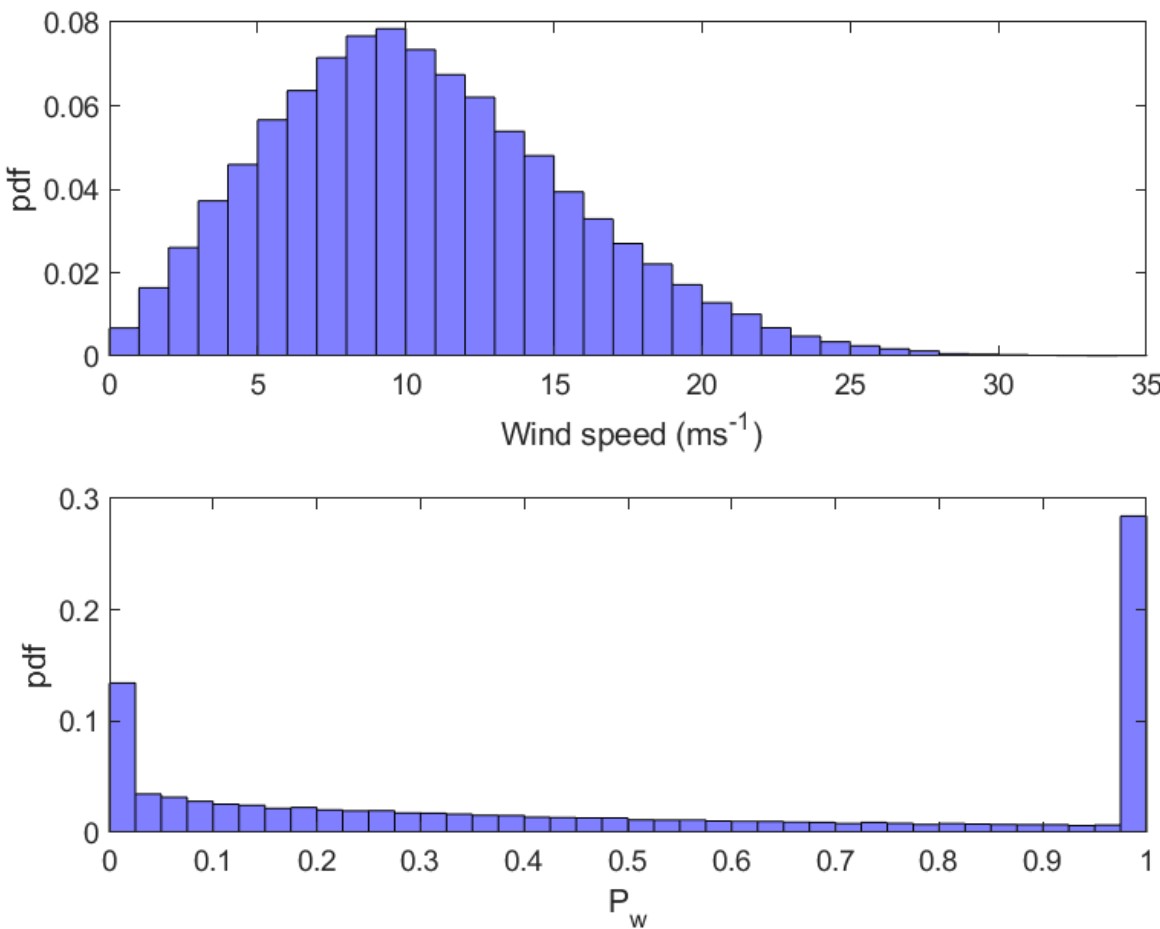

**Figure 2.** Example distribution for Ekofisk (ek) showing the the probability density function (pdf) for both wind speed ($ms^{-1}$) and normalized wind power ($P_w$).

| Platforms | Abbreviation | Wind speed | | | | Wind power | | | |
|---|---|---|---|---|---|---|---|---|---|
| | | $\mu$ | $\sigma$ | a | b | $q_{50}$ | IQR | RCoV | CF |
| Ekofisk | ek | 10.49 | 5.28 | 11.85 | 2.09 | 0.43 | 0.89 | 0.90 | 0.5 |
| Sleipner | sl | 10.86 | 5.89 | 12.25 | 1.92 | 0.47 | 0.90 | 0.95 | 0.52 |
| Gullfaks C | gf | 10.77 | 6.04 | 12.13 | 1.85 | 0.46 | 0.92 | 0.96 | 0.51 |
| Draugen | dr | 9.97 | 5.93 | 11.19 | 1.73 | 0.33 | 0.95 | 1.00 | 0.45 |
| Norne | no | 11.25 | 5.75 | 12.70 | 2.05 | 0.52 | 0.89 | 0.92 | 0.53 |

**Table 2.** Statistical measures of the wind speed and wind power. $\mu$ and $\sigma$ are the Weibull mean and standard deviation, respectively, 'a' and 'b' are the scale and shape parameter of a Weibull distribution, respectively, $q_{50}$ is the median (second quartile), IQR is the inter-quartile range, RCoV is the robust coefficient of variability, and CF is the wind power capacity factor.

narrow) distribution. For this data set "a" ranges from 11.2 to 12.7 for Draugen (dr) and Sleipner (sl), respectively. "b" tells us about the shape of the distribution. A small "b" ($b < 3$) means that the distribution is positively skewed, with a long tail to the right of the mean. The smaller the number, the more right-skewed the distribution. Here, "b" ranges from 1.73 to 2.09 for Draugen (dr) and Ekofisk (ek), respectively, meaning that all the distributions are positively skewed, confirming that the wind speed has more of a Weibull distribution than a Gaussian distribution. Since the goal of a functioning wind park is to produce

as much wind power as possible, it is desirable that the wind speed data fall between $4 < u \le 25$, or even more preferably between $13 < u \le 25$ which will give $P_w = 1$. Sleipner (sl) is the site that most frequently operates at rated power: slightly over 30 % of the time. This is a result of the fact that Sleipner (sl) has an optimal combination of the scale (12.3) and shape (1.9) parameter, with the largest portion of the wind speed distribution falling between $4 \, ms^{-1}$ and $25 \, ms^{-1}$.

For the wind power, the median values ($q_{50}$) range from 0.33 for Draugen (dr) to 0.52 for Norne (no). Another measure of the

155 performance of a wind park is the capacity factor (CF), which is defined as the annual mean power production divided by the installed capacity. Draugen (dr) has the lowest capacity factor, $CF = 0.45$, and Norne (no) has the highest, with $CF = 0.53$. The wind power IQR is high, around 0.9. This means that $q_{50} \pm \frac{IQR}{2}$ contains 50 % of the wind power output. There is therefore potential for reducing wind power intermittency by combining sites.

Reichenberg et al. (2014) concluded that the coefficient of variability for wind power can be substantially reduced by ge-

160 ographic allocation of the production sites. They used the arithmetic mean ($\mu$) and standard deviation ($\sigma$), calculating the coefficient of variability ($CV = \frac{\sigma}{\mu}$). As mentioned above, using the arithmetic mean and standard deviation gives rise to a misleading interpretation of the actual middle value and the accompanying variability of the wind power output. This is due to the very different wind power distribution arising from the non-linear conversion of the wind speed data to wind power output through the power conversion curve (see Eq. 3). Hence, another more robust and resistant measure of wind power variability

is the RCoV (Lee et al., 2018). The $RCoV = \frac{MAD}{q_{50}}$ is the median absolute deviation (MAD) divided by the median ($q_{50}$) and is a normalized measure of the spread in the data set. Here, RCoV ranges from 0.9 for Ekofisk (ek) to 1.00 for Draugen (dr), meaning that a typical deviation from the median value is approximately equal to the median value itself.

### 4.1.1  İnter-annual and seasonal wind power variability

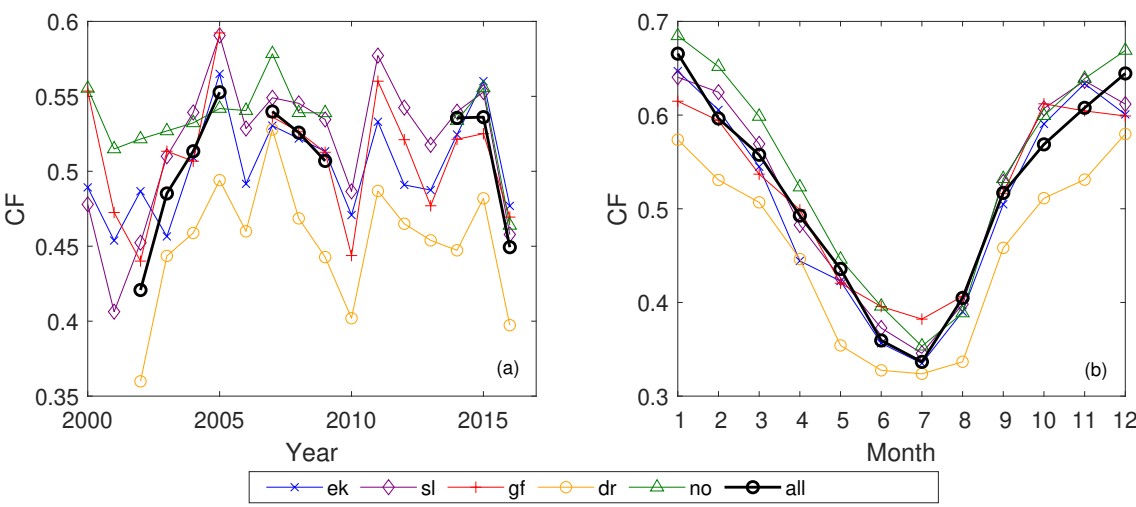

**Figure 3.** (a) Inter-annual variation in the capacity factor (CF) for the five sites and the total interconnected system ("all"). If more than half of the data in one year were missing the value for that year was excluded in the plot. (b) The seasonal variation in CF for the five sites and the interconnected system ("all").

How CF varies from one year to another gives a good indication of the long-term fluctuations in wind power production. The annual variation in CF can be quite substantial and is presented in Fig. 3 (a). For all the sites, the inter-annual variation can change by up to 0.12 (12 % of installed capacity) from one year to the next (i.e. 2010-2011). On this time scale, the CF values for the five stations follow each other more or less, bearing in mind that on annual timescale, the production in the whole region will strongly co-vary. The strong inter-annual variations in CF clearly demonstrate that measuring the wind conditions over too short time period (i.e. one year) is generally not sufficient to estimate a representative wind power potential for a site at these latitudes.

Figure 3 (b) presents the seasonal variation in CF. The highest CF values are found during the autumn and winter (October-February) and range from 0.58 to 0.68 for the northernmost sites Draugen (dr) and Norne (no), respectively. The lowest CF values occur during the summer, and range from 0.32 at Draugen (dr) to 0.38 at Gullfaks C (gf).

### 4.2  Correlation and intermittency

In terms of reducing wind power variability the optimal hypothetical case would have been to combine production from two stations with correlation coefficient $r = -1$. The two combined sites would then be completely out of phase, and the sum of the individual productions would be constant in time, given equally installed capacity at each site. Since the synoptic weather systems constitute the main source of spatio-temporal variance in wind over open ocean, the connected production sites situated

far apart in our site array will probably experience the most contrasting weather and therefore result in the largest dampening of the wind power intermittency. To quantify this we investigated the pair-wise correlation between the different wind power time series ($P_w$) as a function of distance and time lag between the hypothetically connected sites. Sinden (2007) anaysed the characteristics of the wind power resource for 66 onshore sites in the United Kingdom. He found a substantial reduction in the correlation between site-pairs at separation distances less than $600 \ km$. However, for wind sites located more than $800$ $km$ apart, the decrease in correlation are less pronounced, despite a further increase in the distance between them. Our results confirm this distance-dependency of the correlation found in Sinden (2007): Fig. 4 illustrates how the correlation between station pairs changes as a function of the separation distance. The correlation drops off quickly as the distance (x) between the sites increases. After $x \approx 800 \ km$ the decrease in correlation with distance is reduced to 0.1 and continues to decrease more slowly with increasing separation distance. It is expected that the correlation between site-pairs will approach zero when the separation distance become large enough, meaning that the wind at these sites are completely independent. Some site-pairs can even have slightly negative correlation. Reaching this slowdown in the relation between the correlation and the separation distance after $x \approx 800 \ km$ indicates that combining sites outside a radius of $x \approx 800 \ km$ for further variability reduction has almost a negligible effect for the length and time scales considered here. Nevertheless, the correlation coefficient never drops to zero, or below zero, over the range of the data covered in this study, indicating that none of these station pairs will either be anticorrelated or have completely independent production ($r \leq 0$).

| | Decorrelation length $L$ | | |
| --- | --- | --- | --- |
| **Season** | $e^{bx^a}$ | $ae^{bx}$ | $ae^{bx}$ |
| Annual | 403.01 | 414.56 | 413.65 |
| Winter | 288.97 | 293.51 | 300.69 |
| Spring | 364.80 | 366.74 | 367.20 |
| Summer | 388.11 | 394.91 | 394.53 |
| Autumn | 385.82 | 388.99 | 389.04 |

**Table 3.** Annual and seasonal decorrelation length $L$ (in km) for the three exponential functions in Fig. 4. The third column represent a fit where the point $[x, y] = [0, 1]$ is added to the data points to include that the correlation = 1 when the $x = 0$.

The decorrelation length illustrates at what radius the wind power correlation drops to a fraction of the initial value at $x = 0$. In our study, we use the e-folding distance[2] as a measure of the decorrelation length $L$. The 10 station-pairs are used to identify a best-fitting curve describing the dependency between correlation and separation distance, which will give a general description of the decorrelation length $L$ (in km). Identifying such a best-fitting curve may be challenging, and we therefore use three exponential functions with slightly different properties to indicate the uncertainty in the estimates due to the choice of fitting function. The exponential curves, together with the 10 correlation points are presented in Fig. 4, while the corresponding decorrelation lengths are presented in Table 3. The decorrelation length $L$ is slightly more than $400 \ km$. St. Martin et al. (2015)

---

[2]The distance where the correlation has dropped to $\frac{1}{e} = 0.37$.

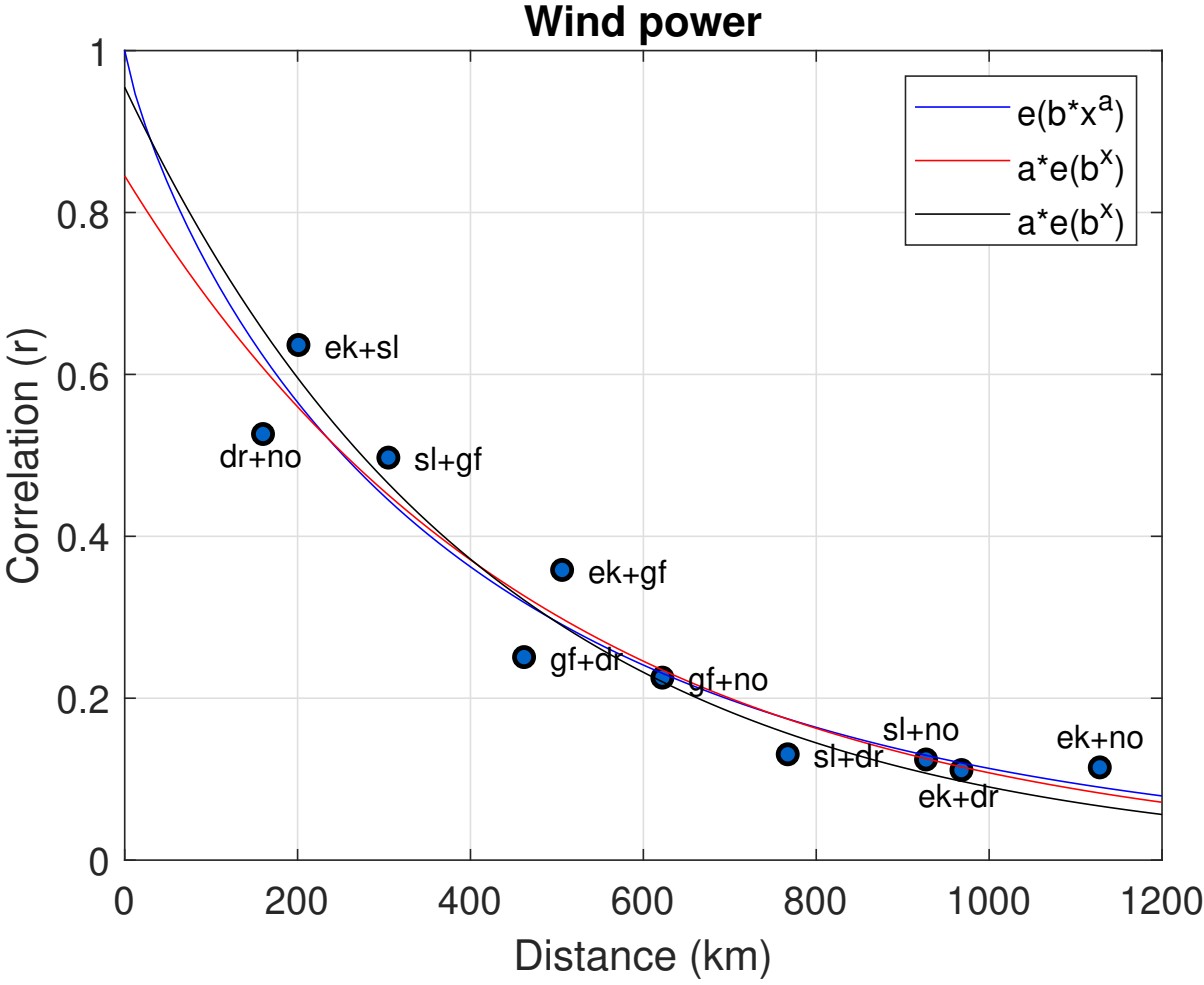

**Figure 4.** How the correlation between site-pairs changes with distance, including three exponential curves giving the best least-squares fit for the 10 correlation-points.

identify decorrelation lengths in the same order using the e-folding distance ($L = 388\ km$, $L = 685\ km$ and $L = 323\ km$ for three different regions: Southeastern Australia, Canada, and Northwestern US, respectively). Further, they argue that more correct decorrelation lengths can be obtained by using the e-folding distance times the nugget effect ($\beta L$), and even better results by using the integral-scale matrix $\xi_\tau$. The integral scale matrix is a measure of the distance required for the correlation to fall to a small value compared to unity. Both measures mentioned above gave a decorrelation length substantially less than the e-folding distance with $\beta L = 273\ km, 447\ km, 130\ km$ and $\xi_\tau = 273\ km, 368\ km, 89\ km$. St. Martin et al. (2015) also concluded that the decorrelation length is highly sensitive to variability time-scale. This result is also obtained in this study (not shown) and in Czisch and Ernst (2001). On time scales longer than a day, St. Martin et al. (2015) found that the benefit of variability-reduction from aggregation of wind power over a region of a given size is independent of time scale. Therefore, if two of our offshore wind producing sites were to balance each other at short time scales ($< 1\ h$), the separation distance could be further reduced from $L = 400\ km$. On the other hand, if they were to balance each other on longer time scales ($> 1\ h$), the separation distance would be larger than $L = 400\ km$. This is a significant result because it underlines the importance of considering time-scales when combining wind power producing sites to reduce wind power intermittency.

In general, the seasonal $L$ is shorter than the annual decorrelation length (see Table 3). The winter months (December-February) contain the shortest decorrelation lengths, varying from $L = 289\ km$ to $301\ km$, depending on the exponential fit. During the winter, the atmosphere is more irregular and chaotic both in space and time, meaning that two stations located a given distance apart will more often enter different wind-regimes during the winter than the other seasons. The summer months (June-August) have the longest decorrelation lengths, spanning from $L = 388\ km$ to $395\ km$. The large-scale atmospheric patters are larger, smoother, more stationary, and last longer. As demonstrated by St. Martin et al. (2015), among others, the decorrelation length is sensitive to the variability time-scale. The decorrelation length increases when the variability time-scale increase. When looking at variability on annual time-scale the seasonal variability also has to be balanced. Hence, the annual decorrelation length is larger than the seasonal decorrelation length.

Figure 5 demonstrates how the time lag of maximum correlation between station pairs changes with distance. In accordance with our expectations, the sites that are closest to each other have the shortest time lag and the highest correlation, implying shorter time for the feature to propagate between them, resulting in a higher correlation. The high- and especially the low-pressure systems that sweep over the North Sea and the Norwegian Sea have a prevailing travelling direction from west to east, due to the strong westerlies at these latitudes. This implies that winds accompanying the system will strike the westernmost site, Sleipner (sl), first, followed by Ekofisk (ek) and Gullfaks (gf) C at short time-intervals, and later Draugen (dr) and Norne (no). A large time lag is beneficial for wind energy production. Then, a wind-event will not occur simultaneously at the both the connected sites. To ensure a time lag approaching $10\ h$, from one site to experience a certain wind-event to the other connected site experiencing the same wind-event, the separation distance needs to exceed $\approx 600 km$.

### 4.3 Connecting wind power sites

To study the effect of interconnected production sites we have to make combined wind power time series for all the different site-configurations. The collective wind power time series, $P_w^c$, for different configurations of sites are found by calculating

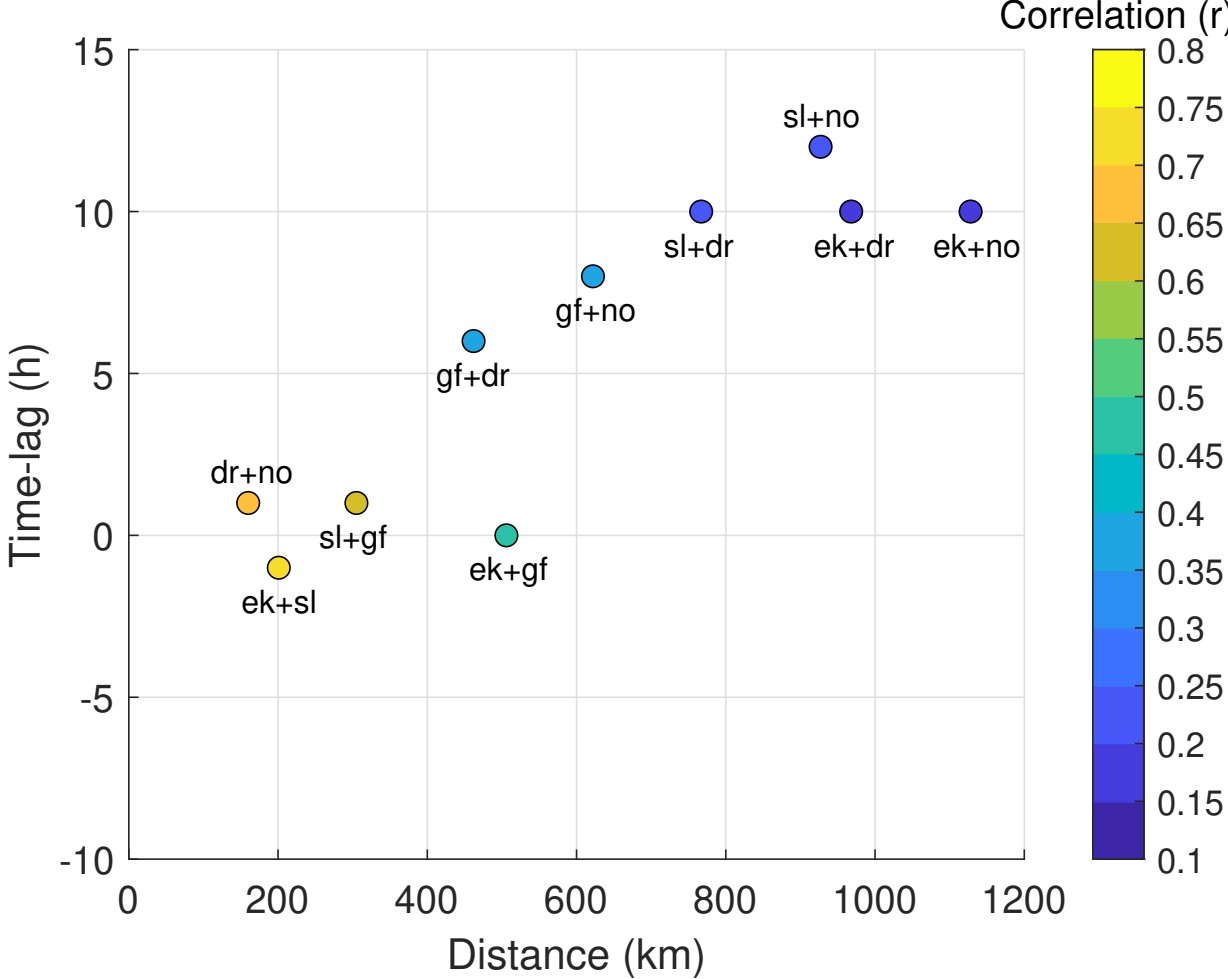

**Figure 5.** Time lag (h) of maximum correlation between the connected sites as a function of the distance between them.

the average power production for each time-step. Time-steps containing NaN-values for any of the sites in the configuration in question are not included. Hence, for a given time step:

$$P_w^c(i) = \frac{1}{j} \sum_{j=1}^{J} P_w^j(i) \tag{4}$$

where i = 1,2,..,N is the timestep, j = 1,2,..,J is the number of sites combined, and $P_w^j(i)$ is the different wind power time series
for an array combination of j sites (1 to 5 sites) at time step i.

For example, a combination of two sites (a and b) at time step i will be:

$$P_w^c(i) = \frac{1}{2} \sum_{j=1}^{2} P_w^2(i) = \frac{1}{2}(P_w^a(i) + P_w^b(i)) \tag{5}$$

This calculation is done for each time step i, as long as $P_i^a \neq NaN$ or $P_i^b \neq NaN$.

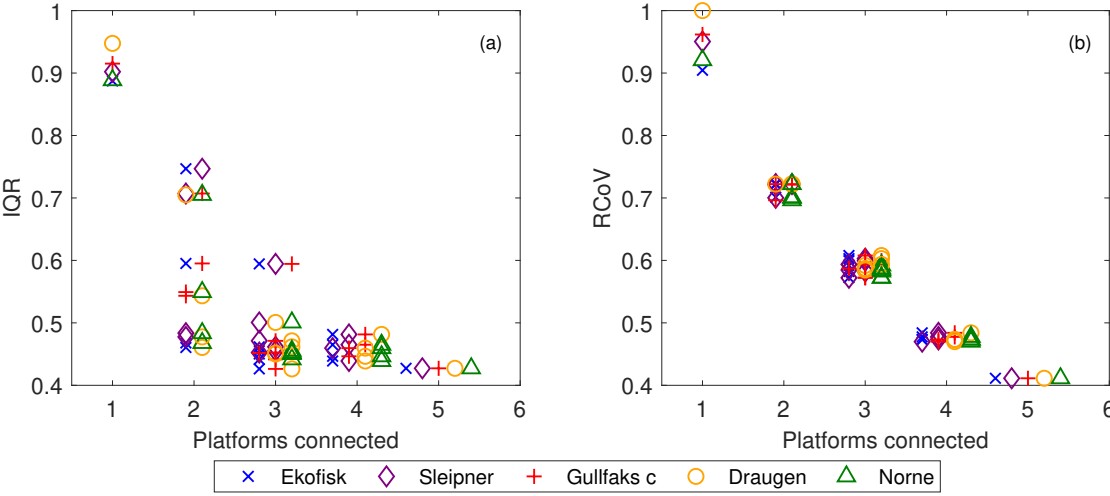

**Figure 6.** (a) and (b) demonstrate how IQR and RCoV change as the array-size of connected sites increases from 1 (single sites) to 5 (all sites connected), respectively.

As mentioned in section 4.1 the IQR is a candidate for estimating wind power variability Kempton et al. (2010). According to Lee et al. (2018) a more robust and resistant variability measure is the robust coefficient of variability. Figure 6 presents IQR and RCoV for different configurations of collective wind power production, $P_w^c$. Common to both the variability measures is that the variability generally decreases quickly with increasing array-size of connected sites. This result clearly demonstrates the advantage of having interconnected wind power production in terms of intermittency reduction: Instead of operating wind turbines at two sites separately, we see that the intermittency of a connected site-pair is reduced, and is further reduced with increased array-size. The site with the highest (lowest) IQR is the station with the lowest (highest) RCoV, since RCoV is normalized by the median value.

A counter-intuitive result is that for some site-combinations the variability (IQR and RCoV) is less than the variability for a larger array-size. For example, the combination of ek+sl+gf has higher variability than most of the pair-wise site-combinations. Some pair-wise combinations even have a lower IQR and RCoV than a four-site combination. This result appears to be as a consequence of the geographic locations of the sites in this study. Since Ekofisk (ek), Sleipner (sl), and Gullfaks C (gf) are roughly aligned in a north-south direction, they will experience the same wind-event more or less simultaneously (see Fig. 5) due to the passage of extra-tropical cyclones and the associated fronts. Therefore, a combination of these sites would be poorer in terms of intermittency reduction than other site combinations, and even combinations of smaller array-size.

### 4.3.1 Wind power generation duration

A typical wind power distribution can be seen in Fig. 2 (lower panel). When connecting sites, the wind power distribution changes shape. As the array size of interconnected sites increases, the distribution converges towards a bell-shaped distribution (Kempton et al., 2010). Unlike the individual distributions, where the most frequent wind power production is $P_w = 1$ followed by $P_w = 0$, the interconnection of production sites results in a production capacity that more regularly falls in the middle of the production range ([0 1]). Nevertheless, it is worth mentioning that when all five sites are combined, the most frequent production mode is still $P_w^c = 1$, indicating that full production is still the most common production state. This result arise since the five sites are located in "superb" wind speed climate. This result can be further discussed in conjunction with the generation duration curve (GDC). Here, the GDC is given as a percentage of the total time the wind power output is above or below a given threshold. As can be seen in Fig. 7, the individual sites produce no power at all between 8 % and 12 % of the time. In contrast to the interconnected system ("all") that almost never experiences zero wind power production. The less steep curve from the interconnected system indicates a less fluctuating output, with a production that more often falls at values near the median value.

## 4.4 Critical power events

The long record of observations (16 years) enables us to make estimates of the risk of having critically low wind power production. In this study, we have chosen to examine the time fraction the wind power production is zero ($P_w = 0$ or $P_w^c = 0$).

The risk ($R_i$) of zero wind power is the sum of the risk of having too low ($R_i^{low}$) and too high ($R_i^{high}$) wind speed, $R_i = R_i^{low} + R_i^{high}$. $R_i$ is calculated by taking the sum of all the hours the power is zero divided by the total number of time steps (NaN-values are not included) and is calculated in the following way:

$$R_i = \frac{1}{n}\sum_{i=1}^{n} i\delta, \qquad \delta = \begin{cases} 1 & if\ u < 4 \wedge u \geq 25 \\ 0 & else \end{cases} \tag{6}$$

where i is the time step and $\delta$ is a modifier and takes on the values 0 or 1, depending on the wind speed $u$. This formula is valid for both the individual wind power time series and the collective wind power time series (see section 4).

A critical question, given a pair-wise connection of sites, is how much wind power one of the site produces when the other site is not producing any at all. Figure 8 presents the median wind power production at one site when the wind speed at the other station is causing zero power production. As the distance between two connected sites increases, the median production at the producing site increases. To ensure a median wind power production of at least 25% of installed capacity at one site when the connected site is not producing any power at all, the separation distance needs to exceed $\approx 600\ km$ for the investigated region.

Since wind is a fluctuating physical parameter, the risk of having a wind speed less than a given, low threshold value is rather high. In addition to the well-known smoothing effect achieved by geographic allocation of wind power, Reichenberg et al. (2017) also investigated the resulting impact on low power events. They found that wind power production below 15%

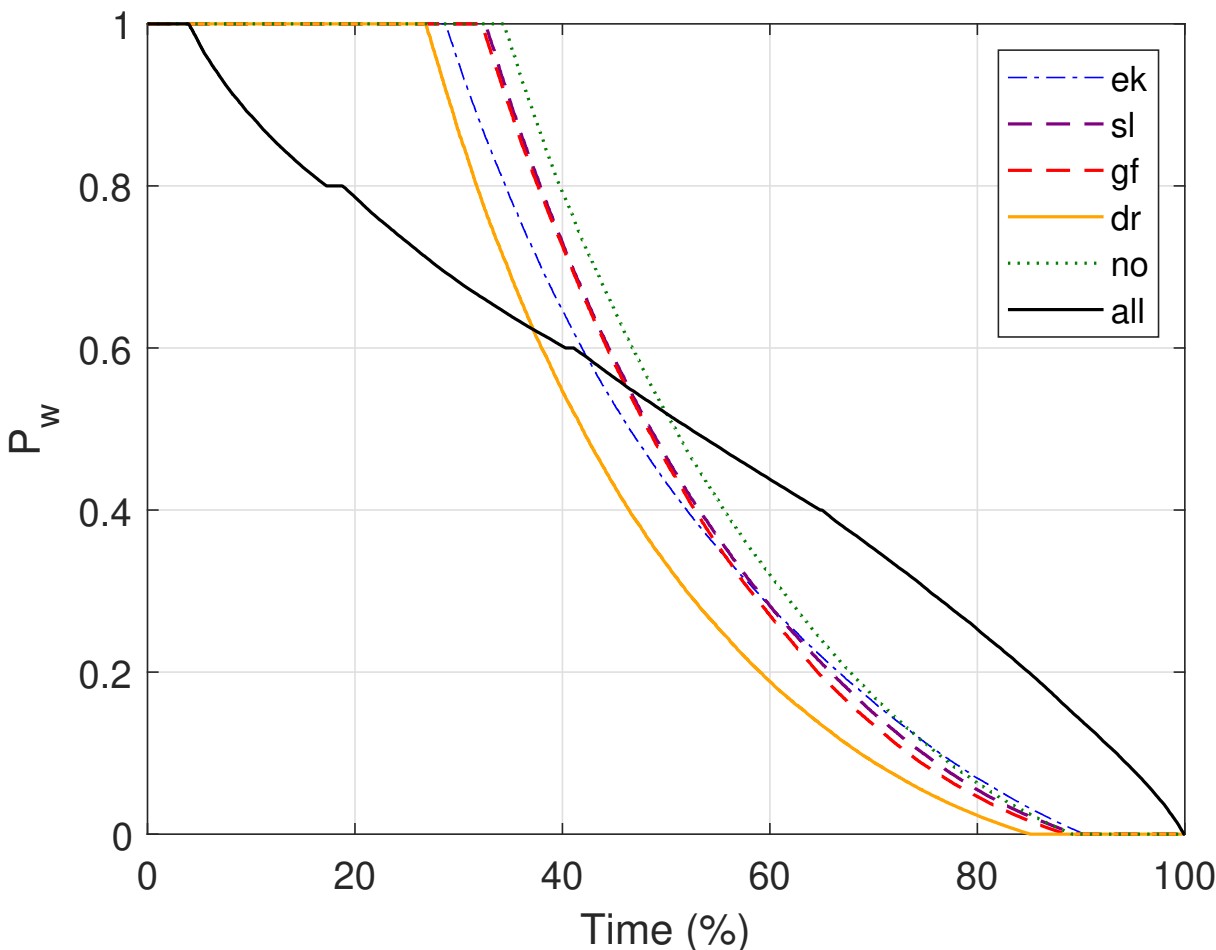

**Figure 7.** Generation duration curve (GDC) for the five sites and the interconnected system ("all").

of installed capacity was hardly ever observed when sites were combined. In contrast to Reichenberg et al. (2017), we study the effect of interconnecting sites on unwanted zero-events. However, the effect is similar to that found by Reichenberg et al. (2017), namely that the occurrence of unwanted events is reduced. The left panel of Fig 9 illustrates how the risk of zero-events changes with distance between the sites and with array-size of connected sites. Going from a single production site to two sites the risk of zero power drops dramatically: from a risk of 8-11 % for a single site to a risk of less than 4 % for two connected sites. Note that the site-combination of Draugen (dr) and Norne (no) has a small risk of $P_w^c = 0$, despite the short distance between between them ($160\ km$). The reason is probably the high wind power production at Norne (no), caused by the topographic effect arising from wind-interactions with the topography in Norway. Hence, the high wind power at Norne (no) compensates for the relatively low wind power production at Draugen (dr) (see Table 2 for median $P_w$ and CF values). This is

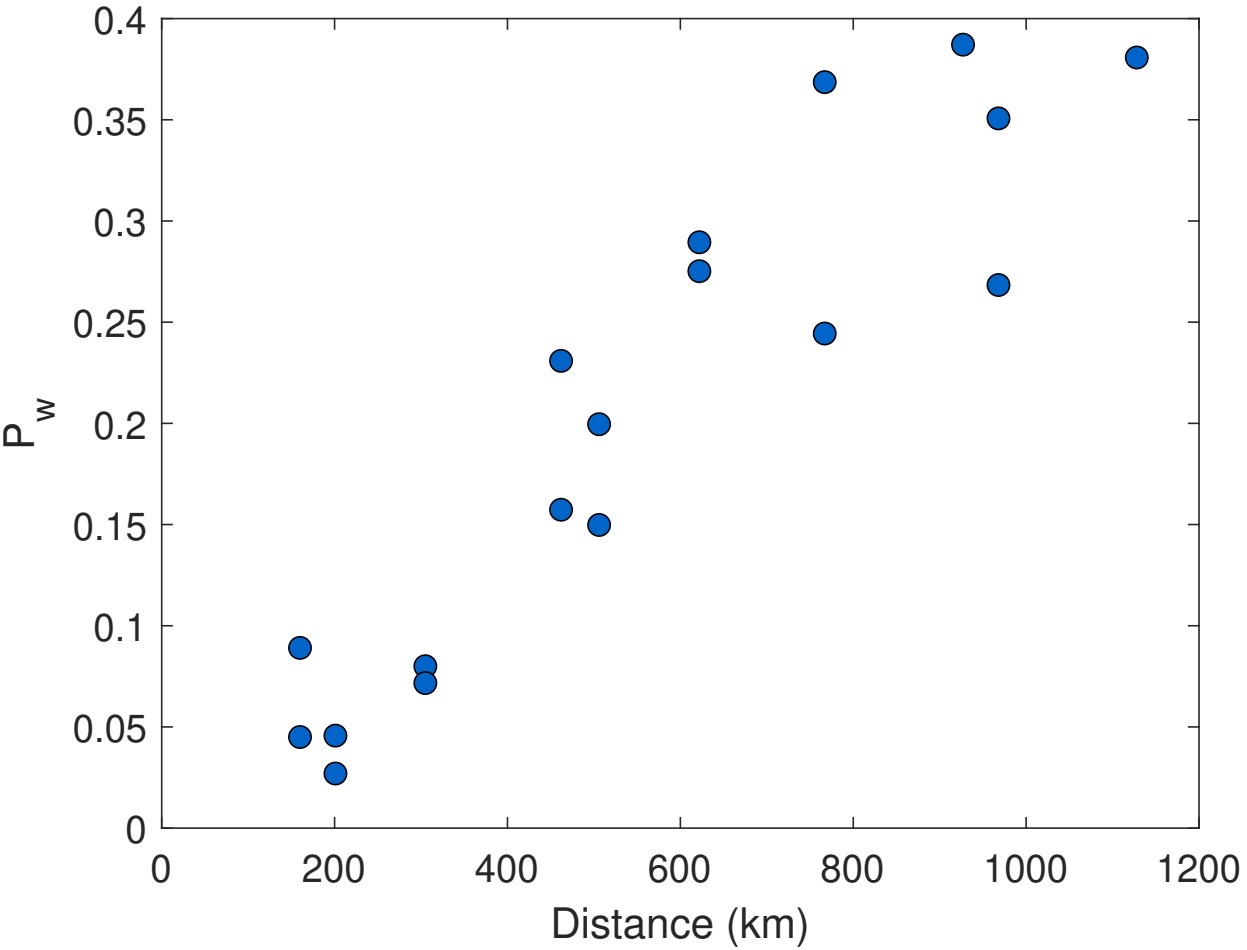

**Figure 8.** Given a pair-vise connection, this figure shows the median $P_w$-production for a site given that the other site are not producing at all ($P_w = 0$) as a function of the distance between the connected site-pair.

relevant for cost estimates related to interconnection. This result also underlines the need for careful selection when connecting
neighboring sites in terms of intermittency reduction. Dvorak et al. (2012) also found that by connecting four offshore wind
farms the occurrence of zero-events was reduced from 9 % to 4 %. By contrast, when we connect four of our sites, the risk of
having $P_w = 0$ is less than 0.5 %. The significant reduction of the risk in this study is due to the greater separation distance
between the sites.

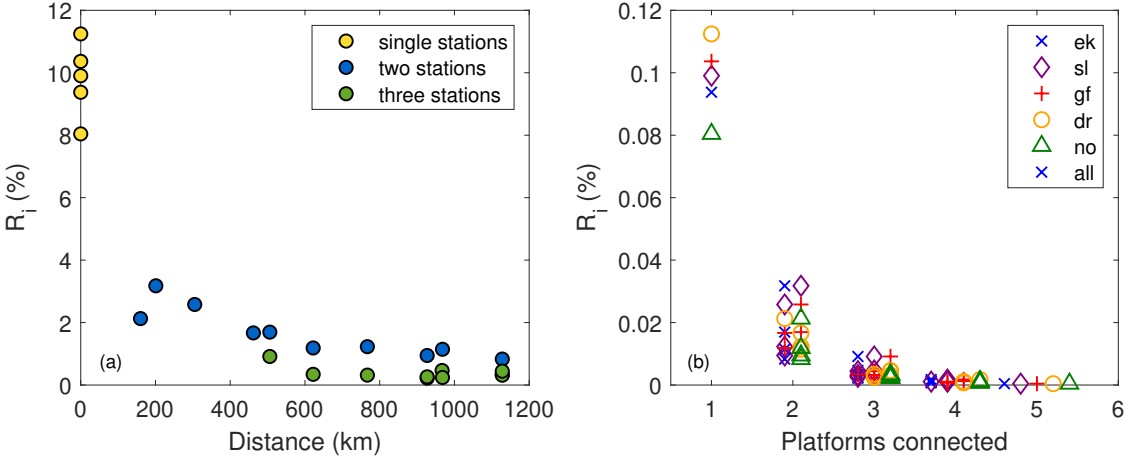

**Figure 9.** (a) demonstrates how the risk ($R_i$) changes with distance between the connected sites. (b) illustrates how the risk ($R_i$) changes as
the array-size of connected sites increases from 0 (single sites) to 5 (all sites connected).

An even more detailed view of the risk of having $P_w^c = 0$ can be seen in the right panel of Fig. 9. This figure tells us which
site-configuration that has the lowest risk of having $P_w^c = 0$. As can be seen in the upper panel, the largest risk reduction is
achieved when shifting from individual site production to a combined two-site production. Increasing the array-size further
reduces the risk, but the reduction is smaller. The configurations with an array-size of three or more have a risk of less than
0.5 % (except the combination of Ekofisk (ek), Sleipner (sl) and Gullfaks C (gf) which has a risk of $\approx$ 1 %). This indicates
that increasing the array-size beyond three might not be financially sound. The fact that the intermittency reduction ceases
is in accordance with the result obtained by Katzenstein et al. (2010). They found that at a frequency of $1\ h^{-1}$ the high- to
low-frequency variability was reduced by 87 % when combining four sites, compared to a single production site, and that
increasing the array-size by the remaining 16 sites resulted in a further intermittency reduction of only 8 %.

### 4.4.1 Extracting wind power during storms

During a strong low pressure system the wind speed can reach wind speed well above the typical cut-out limit of a wind turbine
of $25\ ms^{-1}$. To extract the associated wind power would greatly enhance the full load hours. In reality, present day technology
operates with a turbine shut-down when the wind speed become too strong to prevent damage and destruction. This is referred

to as the well-known "storm control". However, new technology allow the turbines to operates at wind speed exceeding the usual cut-off limit. Instead of an abrupt shut-down of the power extraction at the old cut-out limit, the idea is to introduce a linear reduction of the extraction of wind energy from the old cut-off limit (usually at $25\ ms^{-1}$) to a new and higher cut-out

limit (i.e $30\ ms^{-1}$), here called linear storm control. The difference in wind power production using abrupt power shut down at the old cut-out limit and using linear storm control is showed in Table 4.

| Station | Median | | | Zero power | | |
|---|---|---|---|---|---|---|
| | ST | LST | diff (%) | ST | LST | diff |
| Ekofisk (ek) | 0.43 | 0.45 | **+2.37** | 9.35 | 8.70 | **-0.65** |
| Sleipner (sl) | 0.47 | 0.49 | **+4.29** | 9.9 | 8.84 | **-1.06** |
| Gullfaks C (gf) | 0.46 | 0.48 | **+3.98** | 10.37 | 9.27 | **-1.1** |
| Draugen (dr) | 0.33 | 0.35 | **+4.84** | 11.24 | 10.44 | **-0.8** |
| Norne (no) | 0.52 | 0.54 | **+3.69** | 8.04 | 7.03 | **-1.01** |

**Table 4.** How much the median wind power and the risk of no production changes when a linear storm control is introduced in the power conversion function (see Eq. 3). The storm-control is a linear reduction from the old cut-out limit ($25\ ms^{-1}$) to a new and higher cut-out limit ($30\ ms^{-1}$). ST and LST corresponds to "storm control" and "linear storm control", respectively. The median "diff" is the change in percentage, while the zero power "diff" is the difference in percentage points when introducing linear storm control.

The table show that the median wind power production increases with several percent when introducing linear storm control. From 2.37 % for Ekofisk (ek) to 4.84 % for Draugen (dr), which is quite substantial. On the other hand, for all the sites the risk of having a zero power-event is reduced when introducing the linear storm ontrol. The difference, in percentage points, ranges

from 0.65-1.1. This result indicates that by introducing a linear storm control the turbine will produce more wind power and experience less events of zero-power.

### 4.4.2  Wind power sensitivity related to the power law exponent

The vertical structure of the atmosphere is of major importance when dealing with wind power extraction. How the vertical wind profile looks like depends on the background wind speed, atmospheric stability and the roughness of the surface. As an

approximation, we can estimate the vertical wind speed profile by extrapolation of the wind speed at height z1 to z2 (see Eq. 1).

The atmospheric stability varies from day to day, and even throughout the day. The relation between the power law exponent $\alpha$ and atmospheric stability gives an increase in $\alpha$ with increasing stability. The surface roughness over calm ocean is very low. However, due to the frequent passage of extratropical cyclones at the latitudes in question the ocean surface is often

characterized by large swells and smaller wind driven waves. An increasing surface roughness also increase the value of $\alpha$.

Determining and using a correct $\alpha$-value when mapping the wind power potential is very important, but also demanding. Therefore, a sensitivity analysis of the wind power dependency on the $\alpha$-value is conducted. In Table 5 the median wind power and the risk of zero wind power production for varying power law exponent are listed.

| Station | Median | | | Zero power (%) | | |
|---|---|---|---|---|---|---|
| | $\alpha_l$ | $\alpha_m$ | $\alpha_h$ | $\alpha_l$ | $\alpha_m$ | $\alpha_h$ |
| Ekofisk (ek) | 0.42 | 0.43 | 0.45 | 9.47 | 9.35 | 9.23 |
| Sleipner (sl) | 0.48 | 0.47 | 0.45 | 9.76 | 9.91 | 10.02 |
| Gullfaks C (gf) | 0.48 | 0.46 | 0.44 | 10.29 | 10.37 | 10.50 |
| Draugen (dr) | 0.32 | 0.33 | 0.34 | 11.36 | 11.24 | 11.16 |
| Norne (no) | 0.48 | 0.52 | 0.56 | 8.18 | 8.04 | 7.97 |

**Table 5.** Sensitivity in the median wind power production and the risk of zero power production as a function of the power law exponent $\alpha$. $\alpha_l = 0.08$, $\alpha_m = 0.12$, and $\alpha_h = 0.16$. $\alpha_m$ is the value used throughout the paper.

The wind sensor mounted on each platform are located in different heights, some above ("above-hub" = sl and gf) and some below ("below-hub" = ek, dr and no) the hub-height of the SWT-6.0-154 turbine ($100\ m.a.s.l$). As can be seen from Table 5 choosing a wrong $\alpha$-value to modify the vertical wind speed profile influence both the median wind power production and the risk of zero power production. Using a too low $\alpha$-value in the extrapolation the wind speed for the above- (below-) hub sites will result in a higher (lower) wind power production at $100\ m.a.s.l$ and hence a decreased (increased) risk of having zero wind power production. Vice versa if $\alpha$ takes on a too high value: The wind speed for the above- (below-) hub sites will result in a lower (higher) wind power production at $100\ m.a.s.l$ an increased (decreased) risk of having zero wind power production.

By using the same power law exponent we assume the same state of the ocean surface and also the same atmospheric stability all the five sites. This can of course lead to erroneous wind speed values and hence wind power output. Further investigation in the choice of the power law exponent is outside the scope of this paper.

### 4.4.3 The influence of large structures on the wind field

Oil- and gas platforms are large structures, ranging several tens of meters above the sea surface. The platforms are often located far offshore at areas that are poorly covered in terms of observational data. However, wind sensor mounted on top of these large structures enables us to some extent map the wind conditions at each of these sites.

The wind field over the open ocean is almost undisturbed. However, when the flow is approaching a platform the wind field will start to alter. Several studies have looked at the impact of these large structures on the background flow (ref). A common result is that these large offshore structures impact the wind field. Depending on the wind direction the wind speed is to some extent either accelerated or decelerated by the structure, together with a backing or veering of the wind direction. These structures disturb the background flow causing downwind turbulence to appear.

Using these observations to map the wind characteristics and the associated wind power might give a wrong picture of the actual wind power potential at the site in question. On the other hand, using climate models to produce wind speed climatology for the same site also introduce uncertainties. So, in this study, looking at the effect of interconnection of wind power production sites, we believe that the alteration of the wind power potential by using observations from oil- and gas platforms might be of minor importance.

## 4.5 Zero-events caused by too low or too high wind

Since a zero-event ($P_w = 0$) occurs both when $u < 4\ ms^{-1}$ (too low wind) and $u \geq 25\ ms^{-1}$ (too high wind) we choose to split the zero-power events, when investigating associated meteorological processes. Figure 10 presents both the occurrence and duration of zero-events for two seasons, namely winter and summer, when the most contrasting results occur.

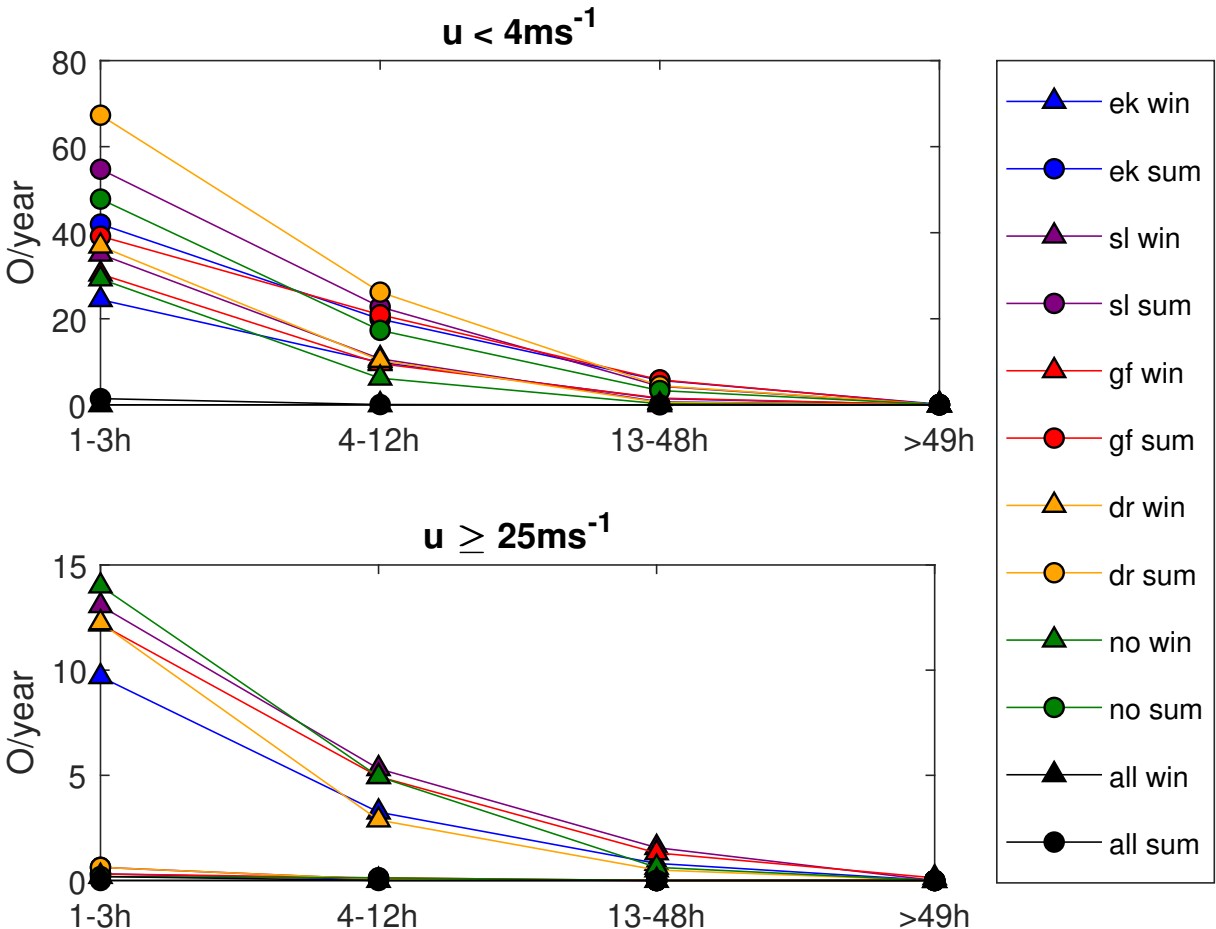

**Figure 10.** Average number of yearly occurrences (O/year) of zero wind power production for different duration (1-3h,4-12h,13-48h and >49h) both for $u \leq 4\ ms^{-1}$ (upper panel) and for $u > 25\ ms^{-1}$ (lower panel). The occurrence of zero power is plotted for winter (triangles) and summer (circles), where the largest differences are seen.

The first thing to notice is that the occurrence of zero-events decreases as the duration increases. In addition, the occurrence of zero-events has almost ceased when all the sites are connected (black curve), and the reduction is most distinct for the shortest duration ($1 - 3\ h$). More zero-events are caused by too low wind (a total of 684.5 yearly zero-events for all the sites)

than too high wind (102.75). The occurrence of zero-events caused by too high wind (too low wind) is highest during winter (summer). In addition, we see that the seasonal difference is largest for the zero-events resulting from too high winds, where the occurrence during the winter is of a larger magnitude than during the summer.

To explain the seasonal differences in the occurrence of zero-events it is necessary to examine the main driving force of variability in the weather phenomena over the open ocean. Synoptic high- and low-pressure systems give rise to the changing weather at our specific sites and are the main contributor to weather-variability over the open ocean. Winds strong enough to terminate power production ($u \geq 25 \ ms^{-1}$) are often associated with the passage of intense low-pressure systems and their accompanying fronts. The occurrence of such strong wind-events is more likely to take place during winter than during summer at the latitudes in question Serreze et al. (1997); Trenberth et al. (1990). Throughout the year, differences in solar insolation give rise to an increased meridional temperature gradient during the northern hemispheric winter. These winter conditions result in a stronger background flow that favors low-pressure activity. On the other hand, the lack of these strong low-pressure systems during summer is probably the main reason why the occurrence of zero-events caused by too low wind is highest during summer. In addition, the fact that blocking high-pressure events are more likely during spring also contributes to the seasonal difference in the too low wind events Rex (1950).

### 4.5.1 The influence of large structures on the wind field

Oil- and gas platforms are large structures, ranging several tens of meters above the sea surface. The platforms are often located far offshore at areas that are poorly covered in terms of observational data. However, wind sensor are mounted on top of these large offshore structures enabling us to some extent map the wind conditions at each of these sites.

The wind field over the open ocean is almost undisturbed. However, when the flow is approaching a platform the wind field will start to alter. Several studies have looked at the impact of these large structures on the background flow (ref). A common result is that these large offshore structures impact the wind field. Depending on the wind direction the wind speed is to some extent either accelerated or decelerated by the structure, together with a backing or veering of the wind direction. These structures disturb the background flow causing downwind turbulence to appear.

Using these observations to map the wind characteristics and the associated wind power might give a wrong picture of the actual wind power potential at the site in question. On the other hand, using climate models to produce wind speed climatology for the same site also introduces uncertainties. So, in this study, looking at the effect of interconnection of wind power production sites, we believe that the alteration of the wind power potential by using observations from oil- and gas platforms might be of minor importance.

### 4.6 Atmospheric conditions causing long-term power shut-down

The previous section demonstrated that the occurrence and duration of zero-events are sensitive to the season and the atmospheric state. Klink (2007) has related long-lasting above- or below-average mean monthly values to variability in selected large-scale atmospheric circulation patterns. This section more closely examines surface pressure patterns associated with zero-events lasting longer than 12 hours using NORA10-reanalysis data (Reistad et al. (2011).

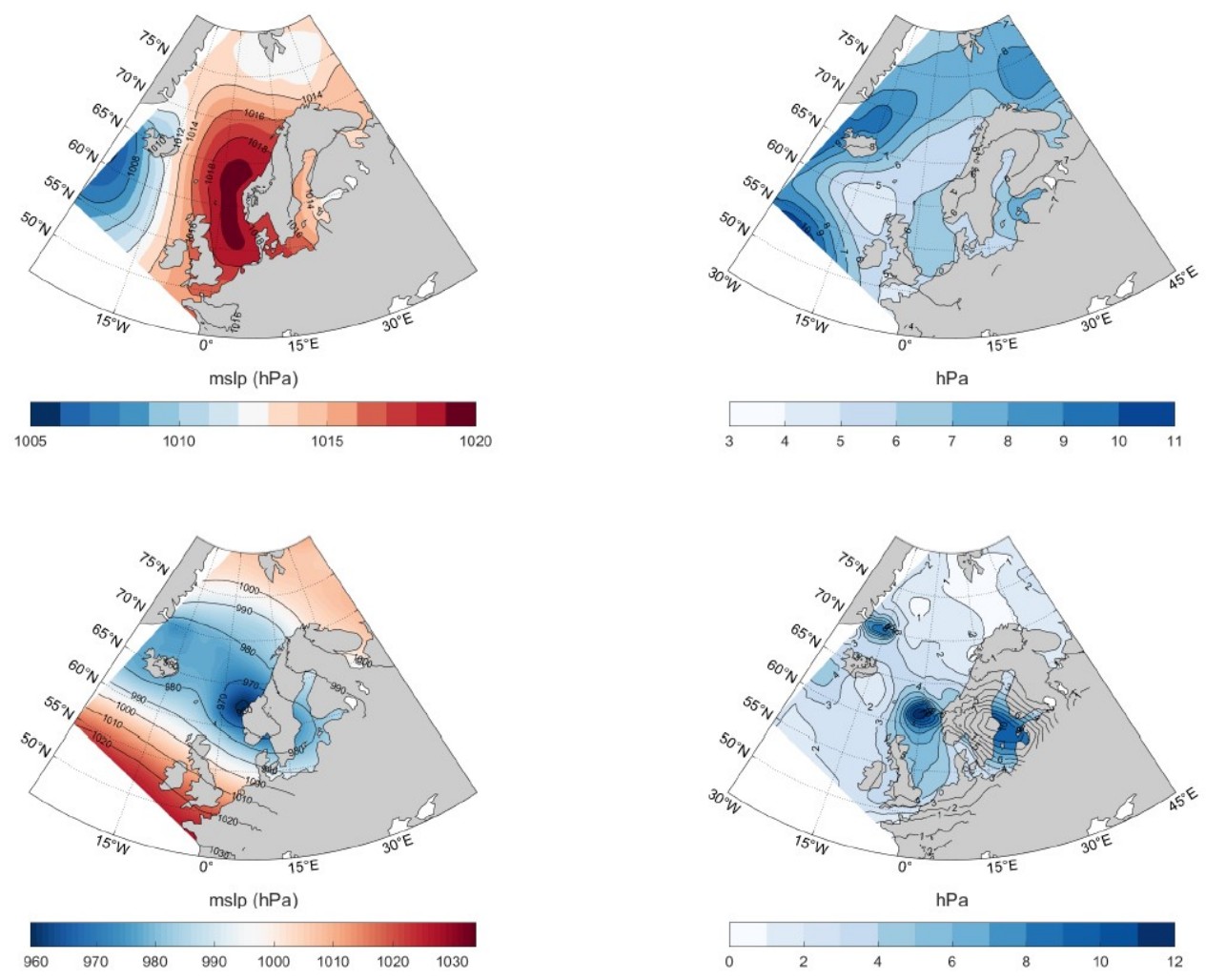

**Figure 11.** Average (composite mean) large scale situations (left column) and the corresponding standard deviations (right column) corresponding to $P_w^c = 0$ for the site-pair Ekofisk+Sleipner (ek+sl). The upper and lower row corresponds to $P_w^c = 0$ caused by too low wind ($u < 4\ ms^{-1}$) and too high wind ($u \geq 25\ ms^{-1}$), respectively.

Surface pressure is a key quantity that contains considerable information about the atmospheric structure in the lower atmosphere. Figure 11 and 12 present the average surface pressure conditions (composite mean) and the corresponding standard deviation associated with zero-events due to too low wind ($u < 4 \ ms^{-1}$) and too strong wind ($u \geq 25 \ ms^{-1}$) for the two site-combinations ek+sl and dr+no, respectively. The average atmospheric condition (comprised of 360 maps) resulting in too low wind speed for ek+sl is a high-pressure system extending from the North Sea and into the Norwegian Sea. This pattern is similar to the positive phase of the Scandinavia pattern (SCAND) Barnston and Livezey (1987). The variability (std) is only a few hPa, indicating that the atmospheric patterns causing too low wind-events are relatively similar to each other, and that most of the variability lies in the patterns´ extension towards the west. By contrast, the average atmospheric situation (comprised of 15 maps) due to too strong wind is an intense low-pressure system hitting the northwest coast of Southern Norway, bringing tight isobars and strong winds over Sleipner (sl) and Ekofisk (ek). This situation is similar to the positive phase of the North Atlantic Oscillation (NAO). The standard deviation is large in the center of the low pressure system. The small spatial extension of the maximum standard deviation indicates uncertainty regarding the depth of the system. However, the std is less over Ekofisk (ek) and Sleipner (sl), indicating that these sites seem to be located to the south of the extratropical cyclone where the strongest winds are often found.

Figure 12 contains the same information as in Fig. 11, but for the site-combination Draugen+Norne (dr+no). The average atmospheric condition (comprised of 159 maps) caused by too low wind is different from that in Fig. 11 (upper left panel). The typical atmospheric condition here is a high-pressure system covering the entire Norwegian Sea and extending across Norway and into Eastern Europe. The std states that the eastern extension of the high pressure system is unclear, giving rise to the large uncertainty east of Norway. Again, as in the case of too high wind for ek+sl, the site-combination dr+no is located to the south of a strong low-pressure system (lower left panel). For dr+no, the mean system is now situated off the coast of northern Norway. The corresponding std is large, indicating that several positions and strengths of the strong low-pressure system can cause situations with too high winds for dr+no.

Even though Kempton et al. (2010) investigated only four specific meteorological situations giving high and low collective wind power, our result are in line with theirs. As for the too low wind events in this study, the too low wind power episodes in Kempton et al. (2010) were associated with a high-pressure system located in the vicinity of the wind power production sites. On the other hand, the episodes resulting in high collective wind power output in Kempton et al. (2010) were characterized by a low-pressure system located in the vicinity of the production sites. This is more or less in line with the results of this paper: our too high wind events are caused by intense low-pressure systems. Accordingly, a less intense low-pressure system will result in high wind power events.

## 5   Summary of main results

In this exploratory study, we quantified the effect of collective offshore wind power production using five sites on the Norwegian continental shelf, which constitutes a unique set of hourly wind speed data observed over a period of 16 years. The sites extend from Ekofisk in the south ($56.5\,^{\circ}$ N) to Sleipner, Gullfaks C, Draugen, and Norne in the north ($66.0\,^{\circ}$ N). See

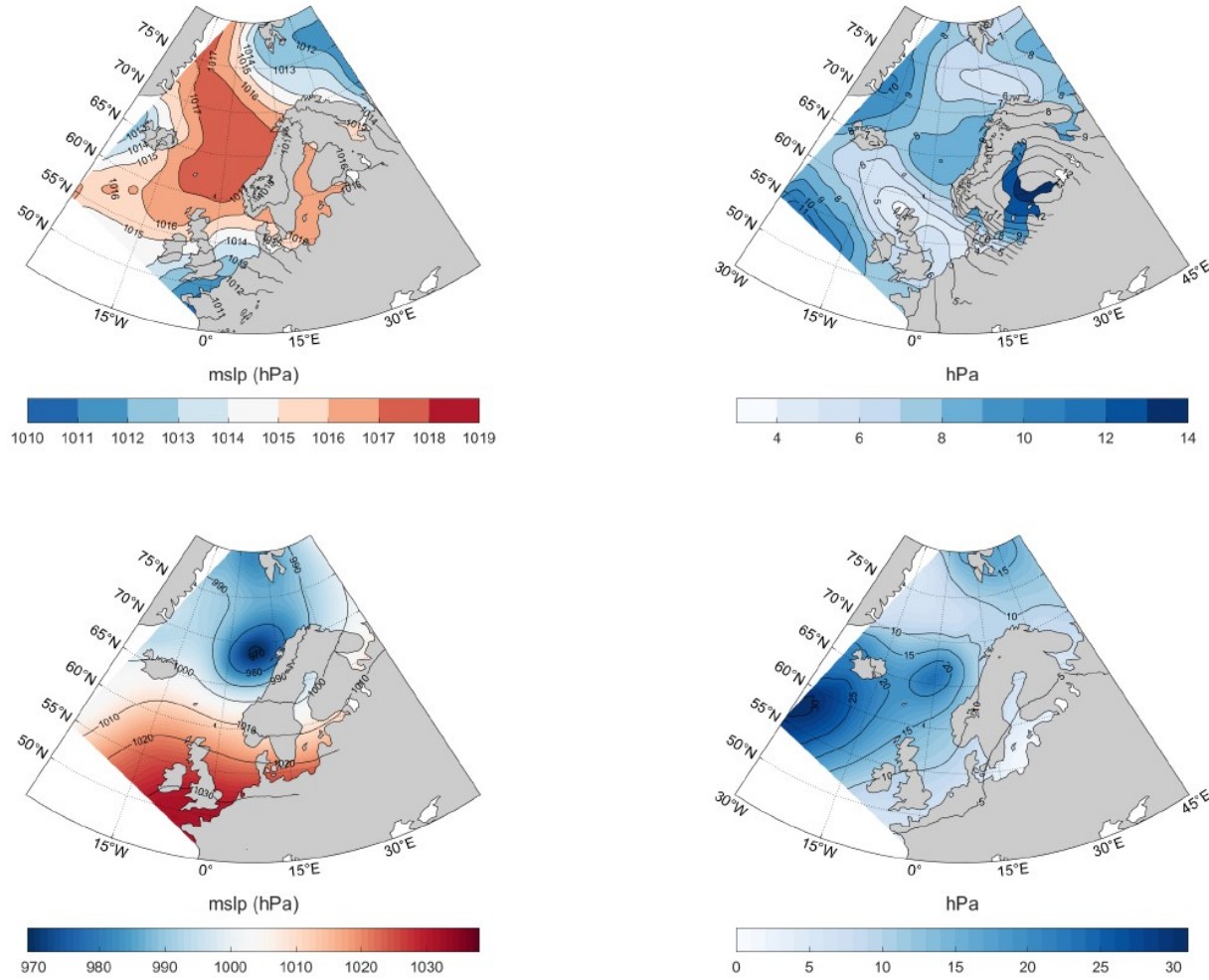

**Figure 12.** Same as in Fig. 11, but for the site-pair Draugen+Norne (dr+no).

Fig. 1 and Table 2 for site details. We addressed the well-known intermittency problem of wind power by means of a hypothetical electricity cable connecting different configurations of the sites. The achieved smoothing effect was quantified by investigating the correlation between the sites as a function of the distance (km) and time-lag (h) between the different pairs of site-combinations. In addition, we studied the potential reduction of critical events (zero wind power-events) for different site-combinations. Moreover, we investigated further details of zero-events grouped into the two categories of: too low and too high wind speed and the corresponding seasonal variations. Finally, the typical atmospheric patterns resulting in zero-events caused by too low and too high wind speed for certain site-combinations were studied. Our main findings are as follows:

– In the case of all five sites, the wind climate was classified as "superb" (category 7), which corresponds to a potential of producing more than $400 \ Wm^{-2}$ of wind energy Zheng et al. (2016). The mean wind speeds at $100 \ m.a.s.l.$ range from $9.97 \ ms^{-1}$ to $11.25 \ ms^{-1}$ for Draugen and Norne, respectively.

– Sleipner is the site that most frequently operates at rated power, 31 % of the time. This is due to the fact that Sleipner has an optimal combination of the scale and shape parameter, with the largest portion of the wind speed distribution falling between $13 \ ms^{-1}$ and $25 \ ms^{-1}$.

– The wind power variability, expressed as IQR and RCoV, ranges from $IQR = 0.89 - 0.95$ for Norne/Ekofisk and Draugen and $RCoV = 0.90 - 1.00$ for Draugen and Norne. Both IQR and RCoV decrease quickly with increasing array-size of connected sites, indicating that wind power intermittency is reduced when sites are connected.

– The pairwise correlation between sites drops off quickly as the distance between the sites increases. However, after $\approx 800 \ km$ the correlation is reduced to 0.1 and continues to decrease more slowly with increasing distance. Reaching this slowdown in the relation between correlation and separation distance after $\approx 800 \ km$ indicates that combining sites farther apart for further variability reduction has almost negligible effect on the length scales possible to explore here.

– The decorrelation length $L$ shows that at a distance $L \approx 400 \ km$ the correlation between site-pairs has dropped to $\frac{1}{e}$. This means that combining sites with at least a decorrelation length apart will substantially reduce wind power intermittency.

– The decorrelation length $L$ increase with variability time-scale. Hence, if two of the offshore wind power-producing sites were to balance each other at shorter time scales ($< 1 \ h$), the separation distance decreases ($L < 400 \ km$).

– To ensure a time-lag of 10 h, from one site to experience a certain wind-event to the other connected site experiencing the same wind-event, the separation distance needs to exceed $\approx 600 \ km$.

– Given a pair-vise site-connection, the separation distance exceeds $\approx 600 \ km$ to ensure a median wind power production of 25 % of installed capacity at one site when the production at the other site is zero.

– The risk of having zero wind power for a given hour decreases from 8.0-11.2 % for the individual sites to less than 4 % when two sites are connected. Increasing the array-size further reduces the risk, but the reduction is smaller.

– The occurrence of zero-events for a given site decreases as the duration increases. Thus, the shorter zero-events (1-3 h and 4-12 h) is more likely to occur than the zero-events lasting longer (more than 13 h).

– For a single site, the total yearly occurrence of zero-events caused by too low wind (high wind) is 684.5 h (102.75 h). By comparison, when all the sites are connected, the total yearly occurrence of zero-events is 1.5 h and 0.2 h for too low and too high wind, respectively.

– The occurrence of zero power-events caused by too high winds (too low winds) is highest during the winter months (summer months). This is due to the increased (decreased) occurrence of strong low-pressure systems at mid-latitudes during the winter (summer).

- The average atmospheric pattern resulting in too strong winds is a low-pressure system located to the north of the combined sites in question. This position of the system leaves the connected pair to the south of the core-center where the strongest winds usually are found in an extratropical cyclone. By contrast, the atmospheric situation resulting in too low winds is a high-pressure system positioned over the connected sites, resulting in very calm wind conditions.

## 5.1 Outlook

This research paper with the working title of "Mitigation of offshore wind power intermittency by interconnection of production sites" has first of all showed that by connecting wind power production sites you reduce unwanted power events, like intermittency and zero-events. The results obtained here are of great importance and leads us to some open questions:

- Given that all locations in the North Sea and the Norwegian Sea could be used for wind power production; where are the best sites for interconnection in terms of a) reducing intermittency, and b) maximizing power output.

- Are the correlation between production sites largest in the East-West or the North-South direction?

- Should the installed capacity at each productions site be unequal in terms of a) reducing intermittency, and b) maximizing power output.

*Author contributions.* Ida Marie Solbrekke: Conceptualization; Data curation; Formal analysis; Investigation; Methodology; Resources; Software; Validation; Visualization; Roles/Writing - original draft; Writing - review & editing

Nils Gunnar Kvamstø: Conseptualisation; Formal analysis; Writing - original draft; Supervision; Funding Acquisition

Asgeir Sorteberg: Formal analysis; Methodology; Supervision; Validation; Visualization; Writing - review & editing.

*Competing interests.* No competing interests to declare.

*Acknowledgements.* Thanks to the Meteorological Institute, and a special thanks to Magnar Reistad, Øyvind Breivik, Ole Johan Aarnes and Hilde Haakenstad for providing all the data used in this study. Thanks to professor David B. Stephenson for discussion of relevant statistics. This work was funded through a PhD grant from Bergen Offshore Wind Center (BOW), University of Bergen.

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
