# Peer review of "Offshore wind power intermittency: The effect of connecting production sites along the Norwegian continental shelf."

_Wind Energy Science, 2020_

## Referee Comment (RC1) · Anonymous Referee #1 · 10 Jul 2020

**General Comment:** The manuscript by Solbrekke et al analyses the impact of connecting different sites that would in principle be suited for (floating) wind power generation on the fluctuation of offshore wind power production. In general, the paper presents novel research, is well written and structured and the figures are clearly readable. However, I have three major comments mainly related to wind turbine technology and the data quality and several minor points. Thus, I recommend publication after dealing with major revisions.

**Major Comments:**

1. The long-term measurement data used in this study is based on wind measure-

ments on oil and gas platforms which are quite massive structures that might quite heavily distort the flow. The manscuript lacks in a description of the expected accuracy of the measurements. A subsection on this should be added to section 2.

2. The authors are interpolating the measured winds to a height of 100m. This is done using the power law. The interpolation in height is quite significant (up to 55 meters). However, the power law has issues in unstable stratification that can e.g. also occur during storms. (see e.g. Emeis 2018). I recommend that a discussion and analysis of the uncertainty this interpolation has on the final results is added to the manuscript before publication as journal paper.

3. The authors assume a full storm shutdown at a cut off wind speed of 25m/s. Recently, several wind turbine manufacturers introduced concepts of a slow shutdown of the turbines in which they operate at a lower level even at wind speeds >25 m/s with a full shutdown in the range of 30m/s in case of the Siemens turbine studied here this is referred to as high wind ride through. The authors should discuss the effect of such an option and at least mention how the occurrence of zero power events would change assuming a gradual reduction of the power in storm conditions.

**Minor Comments:**

1. Title of manuscript: Remove dot and consider reformulation. → I asked myself what "effect" you mean when reading the title for the first time.

2. Abstract – Line 5: "wind power shut-down" → I recommend to add something like due to too high/low winds

3. Line 41 (and throughout the paper): → Space between unit and number and why are units sometimes in italic face and sometimes not?

4. Table 1 and Table 2 (and throughout the paper): → Add a column with the abbreviation. In general I found it a bit difficult to go back and forth between abbreviated and fully spelled names of the stations. I think it is helpful to work with abbreviations everywhere and when the station is spelled with its full name in the text add the abbreviation in brackets.

5. Line 75 – 80: → Are there any physical reasons for the filtering chosen?

6. Figure 1 (caption): sites and the distance (km) between them→ I would add: red lines somewhere

7. Line 106: at nameplate capacity: → Better: rated power

8. Line 107: reaches a cut-out value. . . → I suggest to add the name of what this is (storm shutdown) somewhere here

9. Line 117-118: retrieved from the SWT-6.0-154 → I think it is meaningful to mention here that this turbine was chosen as it is the first commercially operating floating wind farm. Was it?

10. Line 121-123: → I think this can be shortened. A reader of WESC knows that wind speeds are typically Weibull distributed.

11. Figure 2 (caption): → Add abbreviation (ek) to the caption (see minor comment 4 as well)

12. Line 181: Sinden (2007) found a: → I think it is important to mention here where they did these investigations as there might be differences between low/high latitudes etc.

13. Line 190 – cover → covered

14. Line 214: During winter. . .. → I think it would be good to add somewhere in the manscuript what a typical size of a high / low pressure system is because that is driving the distances isn't it?

15. Line 261: Nevertheless it is worth mentioning. . . → I think it should be mentioned here that this is due to the very high "superb" wind speed climate

16. Line 264: produces → produce

17. Line 265: a shut-down in the production → Shut-down sounds so technical to me. Better: production of zero

18. Line 281: separation distance needs to exceed 600 km → add: for the investigated region (see Minor comment 12 as well)

19. Line 288: upper panel of Figure 9 → I guess you mean left panel or better: 9a

20. Line 299: the lower panel of Fig. 9 → I guess you mean right or better: 9b

21. Line 374: the wind speed was classified as "superb" → The wind climate is classified not the wind speed

22. Line 375-376: The mean wind speed range. . . → Add the height (100m) here.

23. Line 377: slightly over 30

24. Line 382: . . .wind power production sites are connected → I think "sites are connected" is sufficient

25. Line 395: . . . from approximately 10

26. Line 397: Thus a short zero-event is more likely to occur than a long-lasting zero-event → What is short and long?

27. Line 399: ... too low wind (high wind) is 684.5 (102.75) ... → What is the unit here? Hours?

28. Line 408: → Somehow, I miss an outlook. What are open questions that could not be solved? Would be interesting to have a short section at the end here.

**References:**

Emeis, 2018: Wind Energy Meteorology, Springer, 2018,

---

## Referee Comment (RC2) · Anonymous Referee #2 · 27 Jul 2020

The paper "The effect of connecting productionsites along the Norwegian continental shelf" byIda Marie Solbrekke et al. investigates the effect of connecting the hypothetical wind power production from up to five different sites located along the norwegian coast. The paper clearly shows how the the fluctuations and the danger of zero power production events can be decreased substantially. The paper is written very well and the result are an important contribution to research in this field.

Regarding the pairwise correlation of the sites the authors conlude that for distances larger than 800km the correlation stops to decrease. In this case the authors should also inlude a fit function into fig. 4 which takes this into account. Can the authors give

a physical explanation for this effect?

---

## Author Comment (AC1) · 21 Aug 2020

**1   General comment to reviewer # 1**

We would like to express our gratitude to referee #1 for his/hers time and effort revising the manuscript with the working title of "Offshore wind power intermittency: The effect of connecting production sites along the Norwegian continental shelf". With the comments from referee 1 we strongly believe that the manuscript now is better and more precise. Please see our response to the comments below (new text added to the paper is in italic).

[Figure]

**2 Respons to major comments from reviewer # 1**

**RC1** The long-term measurement data used in this study is based on wind measurements on oil and gas platforms which are quite massive structures that might quite heavily distort the flow. The manusuript lacks in a description of the expected accuracy of the measurements. A subsection on this should be added to section 2.

**AC1:** Thank you for pointing out the observation-uncertainties introduced when large structures influence the wind field. Please see the subsection (added to the paper) below addressing this issue.

**2.1 The influence of large structures on the wind field**

*Oil- and gas platforms are large structures, ranging several tens of meters above the sea surface. The platforms are often located far offshore at areas that are poorly covered in terms of observational data. However, wind sensor mounted on top of these large structures enables us to some extent map the wind conditions at each of these sites.*

*The wind field over the open ocean is almost undisturbed. However, when the flow is approaching a platform the wind field will start to alter. Several studies have looked at the impact of these large structures on the background flow (ref). A common result is that these large offshore structures impact the wind field. Depending on the wind direction the wind speed is to some extent either accelerated or decelerated by the structure, together with a backing or veering of the wind direction. These structures disturb the background flow causing downwind turbulence to appear.*

*The wind observations used in this study have most likely been influenced by the large structures of oil- and gas platforms, and we do not know to what extent. Using these observations to map the wind characteristics and the associated wind power might give*

*a imprecise picture of the actual wind power potential at the site in question. On the
other hand, using models also introduce uncertainties.*

**RC2:** The authors are interpolating the measured winds to a height of 100m. This
is done using the power law. The interpolation in height is quite significant (up to 55
meters). However, the power law has issues in unstable stratification that can e.g.
also occur during storms. (see e.g. Emeis 2018). I recommend that a discussion and
analysis of the uncertainty this interpolation has on the final results is added to the
manuscript before publication as journal paper.

**AC2:** Thanks for addressing this relevant and important issue regarding the vertical
extrapolation of the wind. A subsection has been written in order to address the wind
power sensitivity on the choice of the power law exponent.

2.2   Wind power sensitivity related to the power law exponent

*The vertical structure of the atmosphere is of major importance when dealing with wind
power extraction. How the vertical wind profile looks like depends on the background
wind speed, atmospheric stability and the roughness of the surface. As an approxima-
tion, we can estimate the vertical wind speed profile by extrapolation of the wind speed
at height z1 to z2 (see Eq. **??**).*

*The atmospheric stability varies from day to day, and even throughout the day. The re-
lation between the power law exponent $\alpha$ and atmospheric stability gives an increase
in $\alpha$ with increasing stability. The surface roughness over calm ocean is very low. How-
ever, due to the frequent passage of extratropical cyclones at the latitudes in question
the ocean surface is often characterized by large swells and smaller wind driven waves.
An increasing surface roughness also increase the value of $\alpha$.*

*Determining and using the correct $\alpha$-value when mapping the wind power potential is
very important, but also demanding. Therefore, a sensitivity analysis of the wind power*

*dependency on the $\alpha$-value is conducted. In Table 1 the median wind power and the risk of zero wind power production for varying power law exponent are listed.*

| | Median | | | Zero power (%) | | |
|---|---|---|---|---|---|---|
| **Station** | $\alpha_l$ | $\alpha_m$ | $\alpha_h$ | $\alpha_l$ | $\alpha_m$ | $\alpha_h$ |
| Ekofisk (ek) | 0.42 | 0.43 | 0.45 | 9.47 | 9.35 | 9.23 |
| Sleipner (sl) | 0.48 | 0.47 | 0.45 | 9.76 | 9.91 | 10.02 |
| Gullfaks C (gf) | 0.48 | 0.46 | 0.44 | 10.29 | 10.37 | 10.50 |
| Draugen (dr) | 0.32 | 0.33 | 0.34 | 11.36 | 11.24 | 11.16 |
| Norne (no) | 0.48 | 0.52 | 0.56 | 8.18 | 8.04 | 7.97 |

**Table 1.** *Sensitivity in the normalized median wind power production and the risk of zero power production as a function of the power law exponent $\alpha$. $\alpha_l = 0.08$, $\alpha_m = 0.12$, and $\alpha_h = 0.16$. $\alpha_m$ is the value used throughout the paper.*

*The wind sensor mounted on each platform are located in different heights, some above ("above-hub" = sl and gf) and some below ("below-hub" = ek, dr and no) the hub-height of the SWT-6.0-154 turbine ($100\ m.a.s.l$). As can be seen from Table 1 choosing a wrong $\alpha$-value to modify the vertical wind speed profile influence both the median wind power production and the risk of zero power production. Using a too low $\alpha$-value in the extrapolation the wind speed for the above- (below-) hub sites will result in a higher (lower) wind power production at $100\ m.a.s.l$ and hence a decreased (increased) risk of having zero wind power production. Vice versa if $\alpha$ takes on a too high value: The wind speed for the above- (below-) hub sites will result in a lower (higher) wind power production at $100\ m.a.s.l$ an increased (decreased) risk of having zero wind power production. As we are lacking information about atmospheric stability, we assume alpha = 0.12 for the rest of the paper. The extrapolation height for Norne is 55m (from the sensor height of $45\ m.a.s.l.$ to $100\ m.a.s.l$) and the expected error due to the extrapolation is highest for this station.*

*There are uncertainties linked to the choice of power law exponent in the wind power extrapolation, and the fact that all the sites were dedicated the same $\alpha$. However,*

*further investigation in the choice of the power law exponent is outside the scope of this paper.*

**RC3:** The authors assume a full storm shutdown at a cut off wind speed of 25m/s. Recently, several wind turbine manufacturers introduced concepts of a slow shutdown of the turbines in which they operate at a lower level even at wind speeds >25 m/s with a full shutdown in the range of 30m/s in case of the Siemens turbine studied here this is referred to as high wind ride through. The authors should discuss the effect of such an option and at least mention how the occurrence of zero power events would change assuming a gradual reduction of the power in storm conditions.

**AC3:** Thank you for the good suggestion of adding an analysis and discussion on the flow distortion due to the platform itself. I have written a subsection. Please see the comment below.

2.3   Extracting wind power during storms

*During a strong low pressure system the wind speed can exceeds values well above the typical cut-out limit of a wind turbine of $25\ ms^{-1}$. To extract the associated wind power would greatly enhance the full load hours. In reality, present day technology operates with a turbine shut-down when the wind speed become too strong to prevent damage and destruction. This is referred to as the well-known "storm control". However, new technology allow the turbines to operates at wind speed exceeding the usual cut-off limit. Instead of an abrupt shut-down of the power extraction at the old cut-out limit, the idea is to introduce a linear reduction of the extraction of wind energy from the old cut-off limit (usually at $25\ ms^{-1}$) to a new and higher cut-out limit (i.e $30\ ms^{-1}$), here called linear storm control. The difference in wind power production using abrupt power shut down at the old cut-out limit and using linear storm control is showed in Table 2.*

*The table show that the median wind power production increases with several percent*

| Station | Median | | | Zero power | | |
|---|---|---|---|---|---|---|
| | ST | LST | diff (%) | ST | LST | diff |
| Ekofisk (ek) | 0.43 | 0.45 | **+2.4** | 9.4 | 8.7 | **-0.7** |
| Sleipner (sl) | 0.47 | 0.49 | **+4.3** | 9.9 | 8.8 | **-1.1** |
| Gullfaks C (gf) | 0.46 | 0.48 | **+4.0** | 10.4 | 9.3 | **-1.1** |
| Draugen (dr) | 0.33 | 0.35 | **+4.8** | 11.2 | 10.4 | **-0.8** |
| Norne (no) | 0.52 | 0.54 | **+3.7** | 8.0 | 7.0 | **-1.0** |

**Table 2.** *How much the median wind power and the risk of no production changes when a linear storm control is introduced in the power conversion function (see Eq. ??). The storm-control is a linear reduction from the old cut-out limit ($25\ ms^{-1}$) to a new and higher cut-out limit ($30\ ms^{-1}$). ST and LST corresponds to "storm control" and "linear storm control", respectively. The median "diff" is the change in percentage, while the zero power "diff" is the difference in percentage points when introducing linear storm control.*

*when introducing linear storm control. From 2.4 % for Ekofisk (ek) to 4.8 % for Draugen (dr), which is quite substantial. On the other hand, for all the sites the risk of having a zero power-event is reduced when introducing the linear storm ontrol. The difference, in percentage points, ranges from 0.65-1.1. This result indicates that by introducing a linear storm control the turbine will produce more wind power and fewer events of zero-power will occur.*

**3   Response to minor comments from reviewer # 1**

**RC1**: Title of manuscript: Remove dot and consider reformulation. I asked myself what "effect" you mean when reading the title for the first time.

**AC1:**: Thanks for the clarifications. The title is now changed to: "Mitigation of offshore wind power intermittency by interconnection of production sites"

**RC2:** Abstract – Line 5: "wind power shut-down". I recommend to add something like due to too high/low winds

**AC2:** Done.

**RC3:** Line 41 (and throughout the paper): Space between unit and number and why are units sometimes in italic face and sometimes not?

**AC3:** Thanks for noticing. The space between unit and number is now added. Now, all the numbers and units are italic (because they are written like equations they turn out italic).

**RC4:** Table 1 and Table 2 (and throughout the paper): Add a column with the abbreviation. In general I found it a bit difficult to go back and forth between abbreviated and fully spelled names of the stations. I think it is helpful to work with abbreviations everywhere and when the station is spelled with its full name in the text add the abbreviation in brackets.

**AC4:** I have now added a column to Table 1 and 2 with the abbreviations. Since the full names of the sites are spelled out quite often throughout the paper, I have added the abbreviations in brackets at some places, just as a reminder.

**RC5:** Line 75 – 80: Are there any physical reasons for the filtering chosen?

**AC5:** The quality criterias do not have a physical reasoning and is set after visual inspection of the time series. They prevent clear outliers and suspicious values. However, we can not rule out that some valid values are filtered out by this approach.

In addition, when I write "Observations with a wind speed tendency $\frac{\delta u}{\delta t} \geq 15 \ ms^{-1}$ over each of two consecutive hours" I mean a spike in the wind speed time series. That is now added to the paper.

**RC6:** Figure 1 (caption): sites and the distance (km) between them! I would add: red lines somewhere

**AC6:** Done.

**RC7:** Line 106: at nameplate capacity: Better: rated power

**AC7:** Done.

**RC8:** Line 107: reaches a cut-out value: I suggest to add the name of what this is (storm shutdown) somewhere here

**AC8:** Good idea. The termination of power when the wind becomes too strong is usually called "storm control". That is added to line 107.

**RC9:** Line 117-118: retrieved from the SWT-6.0-154. I think it is meaningful to mention here that this turbine was chosen as it is the first commercially operating floating wind farm. Was it?

**AC9:** A good point! You are correct - thank you. That is now added.

**RC10:** Line 121-123: I think this can be shortened. A reader of WESC knows that wind speeds are typically Weibull distributed.

**AC10:** Done.

**RC11:** Figure 2 (caption): Add abbreviation (ek) to the caption (see minor comment 4 as well)

**AC11:** Done.

**RC12:** Line 181: Sinden (2007) found a: ! I think it is important to mention here where they did these investigations as there might be differences between low/high latitudes etc

**AC12:** Good point. I have now added the study area in Sinden (2007).

**RC13:** Line 190  cover to covered

**AC13:** Done.

**RC14:** Line 214: During winter: I think it would be good to add somewhere in the manscuript what a typical size of a high / low pressure system is because that is driving the distances isn't it?

**AC14:** Yes, that is mainly the reason for the spatio-temporal variability at these latitudes.

**RC15:** Line 261: Nevertheless it is worth mentioning: I think it should be mentioned here that this is due to the very high "superb" wind speed climate

**AC15:** Done.

**RC16:** Line 264: produces to produce

**AC16:** Done.

**RC17:** Line 265: a shut-down in the production → Shut-down sounds so technical to me. Better: production of zero

**AC17:** The sentence: "In contrast to the interconnected system ("all") that almost never has a shut-down in the production." Is now changes to: "In contrast to the interconnected system ("all") that almost never experiences zero wind power production."

**RC18:** Line 281: separation distance needs to exceed 600 km → add: for the investigated region (see Minor comment 12 as well)

**AC18:** Done.

**RC19:** Line 288: upper panel of Figure 9 → I guess you mean left panel or better: 9a

**AC19:** Yes, thank you. I have now changed it to "left panel of Figure 9".

**RC20:** Line 299: the lower panel of Fig. 9 $rightarrow$ I guess you mean right or better: 9b

**AC20:** Yes, thank you. I have now changed it to "right panel of Figure 9".

**RC21:** Line 374: the wind speed was classified as "superb" → The wind climate is classified not the wind speed

**AC21:** Thank you. That is now changed.

**RC22:** Line 375-376: The mean wind speed range: → Add the height (100m) here.

**AC22:** Done

**RC23:** Line 377: slightly over 30

**AC23:** Replaced "slightly over 30" with the exact value (31).

**RC24:** Line 382: wind power production sites are connected → I think "sites are connected" is sufficient

**AC24:** Done.

**RC25:** Line 395: from approximately 10

**AC25:** Changed from "approximately 10" to the actual interval (8.0-11.2).

**RC26:** Line 397: Thus a short zero-event is more likely to occur than a long-lasting zero-event → What is short and long?

**AC26:** Thank you for pointing this out. I have now added the time-interval of the "short zero-events" (1-3 h and 4-12 h) and "long-lasting zero-event" (longer than 13 h).

**RC27:** Line 399: too low wind (high wind) is 684.5 (102.75). → What is the unit here? Hours?

**AC27:** The units of the yearly number of zero-events are "hours". I have now added the unit to the document.

**RC28:** Line 408: → Somehow, I miss an outlook. What are open questions that could not be solved? Would be interesting to have a short section at the end here.

**AC28:** Thank you for pointing this out. We have now written an outlook and added it

the end of the paper. See the section below.

**3.1 Outlook**

*This research paper has first of all showed that by connecting wind power production sites you reduce unwanted power events, like intermittency and zero-events substantially. However, the observational based results obtained here leads us to some open questions:*

- *Given that all locations in the North Sea and the Norwegian Sea could potentially be used for wind power production, where are the best sites for interconnection in terms of reducing intermittency, and maximizing power output?*

- *Are the correlation between production sites substantially varying between different parts of the region?*

- *To what degree would differences in cut-off, cut-in and rated power at connected production sites help further reducing intermittency and maximize power output?*

*Due to the small number of observational sites, these questions can only be properly answered using high resolution numerical wind modelling data that through validation has been found to properly represent spatial co-variability on different timescales.*

---

## Author Comment (AC2) · 21 Aug 2020

**1   General comment to reviewer # 2**

We would like to thank reviewer #2 for his/hers comment on the manuscript with the working title of "Offshore wind power intermittency: The effect of connecting production sites along the Norwegian continental shelf". Please see the response below (new text added to the paper is in italic).

**1.1 Response to comment from reviewer #2**

**RC1:** Regarding the pairwise correlation of the sites the authors conclude that for distances larger than 800km the correlation stops to decrease. In this case the authors should also include a fit function into fig. 4 which takes this into account. Can the authors give a physical explanation for this effect?

**AC1:** Thanks for pointing this out. We agree that there is no reason why the correlation should cease to decrease with increasing separation distance. The section has now been changed to the following: "*Fig. 4 illustrates how the correlation between station pairs changes as a function of the separation distance. The correlation drops off quickly as the distance (x) between the sites increases. After $x \approx 800\ km$ the decrease in correlation with distance is reduced to 0.1 and continues to decrease slowly with increasing separation distance. It is expected that the correlation between site-pairs will approach zero when the separation distance become large enough, meaning that the wind at these sites are completely independent. Some site-pairs can even have slightly negative correlation. Reaching this slowdown in the relation between the correlation and the separation distance after $x \approx 800\ km$ indicates that combining sites outside a radius of $x \approx 800\ km$ for further variability reduction has almost a negligible effect for the length and time scales considered here. Nevertheless, the correlation coefficient never drops to zero, or below zero, over the range of the data covered in this study, indicating that none of these station pairs will either anticorrelated or completely independent production ($r \leq 0$).*"

---

## Author Comment (AC3) · 24 Aug 2020

**1   General comment to reviewer # 1**

We would like to express our gratitude to referee #1 for his/hers time and effort revising the manuscript with the working title of "Offshore wind power intermittency: The effect of connecting production sites along the Norwegian continental shelf". With the comments from referee 1 we strongly believe that the manuscript now is better and more precise. Please see our response to the comments below (new text added to the paper is in italic).

[Figure]

**2 Respons to major comments from reviewer # 1**

**RC1** The long-term measurement data used in this study is based on wind measurements on oil and gas platforms which are quite massive structures that might quite heavily distort the flow. The manusuript lacks in a description of the expected accuracy of the measurements. A subsection on this should be added to section 2.

**AC1:** Thank you for pointing out the observation-uncertainties introduced when large structures influence the wind field. Please see the subsection (added to the paper) below addressing this issue.

**2.1 The influence of large structures on the wind field**

*Oil- and gas platforms are large structures, ranging several tens of meters above the sea surface. The platforms are often located far offshore at areas that are poorly covered in terms of observational data. However, wind sensor mounted on top of these large structures enables us to some extent map the wind conditions at each of these sites.*

*The wind field over the open ocean is almost undisturbed. However, when the flow is approaching a platform the wind field will start to alter. Several studies have looked at the impact of these large structures on the background flow. A common result is that these large offshore structures impact the wind field. Depending on the wind direction the wind speed is to some extent either accelerated or decelerated by the structure, together with a backing or veering of the wind direction. These structures disturb the background flow causing downwind turbulence to appear. However, in the work "Near-Surface marine wind profiles from rawinsonde and NORA10 hindcast" by Furevik & Haakenstad (2012) they looked at the difference between the wind speed measured by the wind sensors mounted at the platform Ekofisk and an ocean weather ship called Polarfront. The slope of the linear regression line showed that there were no systematic*

*difference in the measured wind speed between the rawinsonde and the wind sensor at the site in question. Since the rawinsonde drifts downwind from the platforms/ship immediately after release this result indicates that the influence by the large structures on the flow might be of minor importance.*

*Using the observations from the platforms to map the wind characteristics and the associated wind power might give an imprecise picture of the actual wind power potential at the site in question. On the other hand, the alternative using wind speeds from numerical weather prediction models would introduce even larger uncertainties.*

**RC2:** The authors are interpolating the measured winds to a height of 100m. This is done using the power law. The interpolation in height is quite significant (up to 55 meters). However, the power law has issues in unstable stratification that can e.g. also occur during storms. (see e.g. Emeis 2018). I recommend that a discussion and analysis of the uncertainty this interpolation has on the final results is added to the manuscript before publication as journal paper.

**AC2:** Thanks for addressing this relevant and important issue regarding the vertical extrapolation of the wind. A subsection has been written in order to address the wind power sensitivity on the choice of the power law exponent.

2.2   Wind power sensitivity related to the power law exponent

*The vertical structure of the atmosphere is of major importance when dealing with wind power extraction. How the vertical wind profile looks like depends on the background wind speed, atmospheric stability and the roughness of the surface. As an approximation, we can estimate the vertical wind speed profile by extrapolation of the wind speed at height z1 to z2.*

*The atmospheric stability varies from day to day, and even throughout the day. The relation between the power law exponent $\alpha$ and atmospheric stability gives an increase*

*in $\alpha$ with increasing stability. The surface roughness over calm ocean is very low. However, due to the frequent passage of extratropical cyclones at the latitudes in question the ocean surface is often characterized by large swells and smaller wind driven waves. An increasing surface roughness also increases the value of $\alpha$.*

*Determining and using the correct $\alpha$-value when mapping the wind power potential is very important, but also demanding. Therefore, a sensitivity analysis of the wind power dependency on the $\alpha$-value is conducted. In Table 1 the median wind power and the risk of zero wind power production for varying power law exponent are listed.*

| heightStation | Median | | | Zero power (%) | | |
|---|---|---|---|---|---|---|
| | $\alpha_l$ | $\alpha_m$ | $\alpha_h$ | $\alpha_l$ | $\alpha_m$ | $\alpha_h$ |
| heightEkofisk (ek) | 0.42 | 0.43 | 0.45 | 9.47 | 9.35 | 9.23 |
| Sleipner (sl) | 0.48 | 0.47 | 0.45 | 9.76 | 9.91 | 10.02 |
| Gullfaks C (gf) | 0.48 | 0.46 | 0.44 | 10.29 | 10.37 | 10.50 |
| Draugen (dr) | 0.32 | 0.33 | 0.34 | 11.36 | 11.24 | 11.16 |
| Norne (no) | 0.48 | 0.52 | 0.56 | 8.18 | 8.04 | 7.97 |
| height | | | | | | |

**Table 1.** *Sensitivity in the normalized median wind power production and the risk of zero power production as a function of the power law exponent $\alpha$. $\alpha_l = 0.08$, $\alpha_m = 0.12$, and $\alpha_h = 0.16$. $\alpha_m$ is the value used throughout the paper.*

*The wind sensor mounted on each platform are located in different heights, some above ("above-hub" = sl and gf) and some below ("below-hub" = ek, dr and no) the hub-height of the SWT-6.0-154 turbine ($100$ m.a.s.l). As can be seen from Table 1 choosing a wrong $\alpha$-value to modify the vertical wind speed profile influence both the median wind power production and the risk of zero power production. Using a too low $\alpha$-value in the extrapolation the wind speed for the above- (below-) hub sites will result in a higher (lower) wind power production at $100$ m.a.s.l and hence a decreased (increased) risk of having zero wind power production. Vice versa if $\alpha$ takes on a too high value: The wind speed for the above- (below-) hub sites will result in a lower*

*(higher) wind power production at $100\ m.a.s.l$ an increased (decreased) risk of having zero wind power production. As we are lacking information about atmospheric stability, we assume alpha = 0.12 for the rest of the paper. The extrapolation height for Norne is 55m (from the sensor height of $45\ m.a.s.l.$ to the hub-height if $100\ m.a.s.l$) and the expected error due to the extrapolation is largest for this station.*

*There are uncertainties linked to the choice of power law exponent in the wind power extrapolation, and the fact that all the sites were dedicated the same $\alpha$. However, further investigation in the choice of the power law exponent is outside the scope of this paper.*

**RC3:** The authors assume a full storm shutdown at a cut off wind speed of 25m/s. Recently, several wind turbine manufacturers introduced concepts of a slow shutdown of the turbines in which they operate at a lower level even at wind speeds >25 m/s with a full shutdown in the range of 30m/s in case of the Siemens turbine studied here this is referred to as high wind ride through. The authors should discuss the effect of such an option and at least mention how the occurrence of zero power events would change assuming a gradual reduction of the power in storm conditions.

**AC3:** Thank you for the good suggestion of adding an analysis and discussion on the flow distortion due to the platform itself. I have written a subsection. Please see the comment below.

**2.3 Extracting wind power during storms**

*During a strong low pressure system the wind speed can exceeds values well above the typical cut-out limit of a wind turbine of $25\ ms^{-1}$. To extract the associated wind power would greatly enhance the full load hours. In reality, present day technology operates with a turbine shut-down when the wind speed become too strong to prevent damage and destruction. This is referred to as the well-known "storm control". However, new*

*technology allow the turbines to operates at wind speed exceeding the usual cut-off limit. Instead of an abrupt shut-down of the power extraction at the old cut-out limit, the idea is to introduce a linear reduction of the extraction of wind energy from the old cut-off limit (usually at $25\ ms^{-1}$) to a new and higher cut-out limit (i.e $30\ ms^{-1}$), here called "linear storm control". The difference in wind power production using abrupt power shut down at the old cut-out limit and using linear storm control is showed in Table 2.*

| | | Median | | | Zero power | | |
|---|---|---|---|---|---|---|---|
| height**Station** | **ST** | **LST** | **diff (%)** | **ST** | **LST** | **diff** |
| heightEkofisk (ek) | 0.43 | 0.45 | **+2.4** | 9.4 | 8.7 | **-0.7** |
| Sleipner (sl) | 0.47 | 0.49 | **+4.3** | 9.9 | 8.8 | **-1.1** |
| Gullfaks C (gf) | 0.46 | 0.48 | **+4.0** | 10.4 | 9.3 | **-1.1** |
| Draugen (dr) | 0.33 | 0.35 | **+4.8** | 11.2 | 10.4 | **-0.8** |
| Norne (no) | 0.52 | 0.54 | **+3.7** | 8.0 | 7.0 | **-1.0** |
| height | | | | | | |

**Table 2.** *How much the median wind power and the risk of no production changes when a linear storm control is introduced in the power conversion function The storm-control is a linear reduction from the old cut-out limit ($25\ ms^{-1}$) to a new and higher cut-out limit ($30\ ms^{-1}$). ST and LST corresponds to "storm control" and "linear storm control", respectively. The median "diff" is the change in percentage, while the zero power "diff" is the difference in percentage points when introducing linear storm control.*

*The table show that the median wind power production increases with several percent when introducing linear storm control. From 2.4 % for Ekofisk (ek) to 4.8 % for Draugen (dr), which is quite substantial. On the other hand, for all the sites the risk of having a zero power-event is reduced when introducing the linear storm ontrol. The difference, in percentage points, ranges from 0.65-1.1. This result indicates that by introducing a linear storm control the turbine will produce more wind power and fewer events of zero-power will occur.*

**3 Response to minor comments from reviewer # 1**

**RC1**: Title of manuscript: Remove dot and consider reformulation. I asked myself what "effect" you mean when reading the title for the first time.

**AC1:**: Thanks for the clarifications. The title is now changed to: "Mitigation of offshore wind power intermittency by interconnection of production sites"

**RC2:** Abstract – Line 5: "wind power shut-down". I recommend to add something like due to too high/low winds

**AC2**: Done.

**RC3:** Line 41 (and throughout the paper): Space between unit and number and why are units sometimes in italic face and sometimes not?

**AC3:** Thanks for noticing. The space between unit and number is now added. Now, all the numbers and units are italic (because they are written like equations they turn out italic).

**RC4:** Table 1 and Table 2 (and throughout the paper): Add a column with the abbreviation. In general I found it a bit difficult to go back and forth between abbreviated and fully spelled names of the stations. I think it is helpful to work with abbreviations everywhere and when the station is spelled with its full name in the text add the abbreviation in brackets.

**AC4:** I have now added a column to Table 1 and 2 with the abbreviations. Since the full names of the sites are spelled out quite often throughout the paper, I have added the abbreviations in brackets at some places, just as a reminder.

**RC5:** Line 75 – 80: Are there any physical reasons for the filtering chosen?

**AC5:** The quality criterias do not have a physical reasoning and is set after visual inspection of the time series. They prevent clear outliers and suspicious values. However, we can not rule out that some valid values are filtered out by this approach.

In addition, when I write "Observations with a wind speed tendency $\frac{\delta u}{\delta t} \geq 15\ ms^{-1}$ over each of two consecutive hours" I mean a spike in the wind speed time series. That is now added to the paper.

**RC6:** Figure 1 (caption): sites and the distance (km) between them! I would add: red lines somewhere

**AC6:** Done.

**RC7:** Line 106: at nameplate capacity: Better: rated power

**AC7:** Done.

**RC8:** Line 107: reaches a cut-out value: I suggest to add the name of what this is (storm shutdown) somewhere here

**AC8:** Good idea. The termination of power when the wind becomes too strong is usually called "storm control". That is added to line 107.

**RC9:** Line 117-118: retrieved from the SWT-6.0-154. I think it is meaningful to mention here that this turbine was chosen as it is the first commercially operating floating wind farm. Was it?

**AC9:** A good point! You are correct - thank you. That is now added.

**RC10:** Line 121-123: I think this can be shortened. A reader of WESC knows that wind speeds are typically Weibull distributed.

**AC10:** Done.

**RC11:** Figure 2 (caption): Add abbreviation (ek) to the caption (see minor comment 4 as well)

**AC11:** Done.

**RC12:** Line 181: Sinden (2007) found a: ! I think it is important to mention here where they did these investigations as there might be differences between low/high latitudes etc

**AC12:** Good point. I have now added the study area in Sinden (2007).

**RC13:** Line 190  cover to covered

**AC13:** Done.

**RC14:** Line 214: During winter:  I think it would be good to add somewhere in the manscuript what a typical size of a high / low pressure system is because that is driving the distances isn't it?

**AC14:** Yes, that is mainly the reason for the spatio-temporal variability at these latitudes.

**RC15:** Line 261: Nevertheless it is worth mentioning: I think it should be mentioned here that this is due to the very high "superb" wind speed climate

**AC15:** Done.

**RC16:** Line 264: produces to produce

**AC16:** Done.

**RC17:** Line 265: a shut-down in the production → Shut-down sounds so technical to me. Better: production of zero

**AC17:** The sentence: "In contrast to the interconnected system ("all") that almost never has a shut-down in the production." Is now changes to: "In contrast to the interconnected system ("all") that almost never experiences zero wind power production."

**RC18:** Line 281: separation distance needs to exceed 600 km → add: for the investigated region (see Minor comment 12 as well)

**AC18:** Done.

**RC19:** Line 288: upper panel of Figure 9 → I guess you mean left panel or better: 9a

**AC19:** Yes, thank you. I have now changed it to "left panel of Figure 9".

**RC20:** Line 299: the lower panel of Fig. 9 $rightarrow$ I guess you mean right or better: 9b

**AC20:** Yes, thank you. I have now changed it to "right panel of Figure 9".

**RC21:** Line 374: the wind speed was classified as "superb" → The wind climate is classified not the wind speed

**AC21:** Thank you. That is now changed.

**RC22:** Line 375-376: The mean wind speed range: → Add the height (100m) here.

**AC22:** Done

**RC23:** Line 377: slightly over 30

**AC23:** Replaced
over 30" with the exact value (31).

**RC24:** Line 382: wind power production sites are connected → I think "sites are connected" is sufficient

**AC24:** Done.

**RC25:** Line 395: from approximately 10

**AC25:** Changed from "approximately 10" to the actual interval (8.0-11.2).

**RC26:** Line 397: Thus a short zero-event is more likely to occur than a long-lasting zero-event → What is short and long?

**AC26:** Thank you for pointing this out. I have now added the time-interval of the "short zero-events" (1-3 h and 4-12 h) and "long-lasting zero-event" (longer than 13 h).

**RC27:** Line 399: too low wind (high wind) is 684.5 (102.75). → What is the unit here? Hours?

**AC27:** The units of the yearly number of zero-events are "hours". I have now added the unit to the document.

**RC28:** Line 408: → Somehow, I miss an outlook. What are open questions that could not be solved? Would be interesting to have a short section at the end here.

**AC28:** Thank you for pointing this out. We have now written an outlook and added it the end of the paper. See the section below.

3.1 Outlook

*This research paper has first of all showed that by connecting wind power production sites you reduce unwanted power events, like intermittency and zero-events, substantially. However, the observational based results obtained here leads us to some open questions:*

- *Given that all locations in the North Sea and the Norwegian Sea could potentially be used for wind power production, where are the best sites for interconnection in terms of reducing intermittency, and maximizing power output?*

- *Are the correlation between production sites substantially varying between different parts of the region?*

- *To what degree would differences in cut-off, cut-in and rated power at connected production sites help further reducing intermittency and maximize power output?*

*Due to the small number of observational sites, these questions can only be properly answered using high resolution numerical wind modelling data that through validation has been found to properly represent spatial co-variability on different timescales.*

---

## Author Response (AR2)

**Response to reviewers**

**Ida M. Solbrekke**

**October 2020**

**1 General comment to reviewer # 1**

We would like to express our gratitude to referee #1 for his/hers time and effort revising our manuscript. Please see our response to the comments below.

**2 Response to minor comments from reviewer # 1**

**RC1**: Page 3 – Line 69: "exploratory study" $\rightarrow$ I suggest to remove the word "exploratory".

**AC1**:: Done

**RC2:** Page 3 – Line 77 (also in the comment) : Observations with a wind speed tendency $\delta u/\delta t \geq 15\ ms^{-1} \rightarrow$ The unit is $m/s^2$ not m/s

**AC2**: Thanks for pointing this out. I have now changes the unit to $ms^{-2}$.

**RC3:**Page 5 – Setting z2 to be the ... z1 to be the $\rightarrow$ That sounds linguistically strange to me (not a native speaker). How about: In our study z2 is the hub height of 100m.a.s.l and z1 is the height of the wind sensors and alpha is 0.12. This way we obtain the....

**AC3:** I agree - thanks. I have now rewritten the sentence in the following way: *In our study z2 is the hub height of* $100\ m.a.s.l$ *and z1 is the height of the wind sensors and* $\alpha = 0.12$. *This way we obtain the extrapolated wind speeds at the hub height for each site.*

**RC4:** Page 8 – Line 150 ... fall between $4 < u \leq 25$... $\rightarrow$ Unit is missing here. The same is true one line below.

**AC4:** Done.

**RC5:** Page 12 – Line 237: to exceed approx. 600km $\rightarrow$ Space between number and unit.

**AC5:** Done.

**RC6:** Page 15 – of at least 25% $\rightarrow$ Add space between number and unit

**AC6:** Done.

**RC7:** Page 20 – Line 359 ... on the background flow (ref) $\rightarrow$ The reference is missing here.

**AC7:** I have now added the correct references.

**RC8:** Page 27 – Line 484-86: This research paper with he working title. . . . → This is a very long sentence, please consider rewriting by e.g. removing the title of the manuscript. Also the "you reduce" sounds very colloquial to me. However, I am not a native speaker.

**AC8:** The sentence is now rewritten as follows: *This research paper has first of all showed that by connecting wind power production sites the unwanted events like intermittency and zero-events are reduced.*

**RC9:** Figure 6: demonstrate how IQR and RCoV change. . . → I suggest to spell out IQR and RCoV out here once again as this helps reading the figure independently of the paper.

**AC9:** Done.

**RC10:** Figure 8: pair-vise → I guess you mean: pairwise

**AC10:** Thanks. Done.

**RC11:** Figure 10: Please add a space between number and unit in the figure title

**AC11:** Done.

**RC12:** Figure 11 and 12: The figure panels are quite small and a lot of space is wasted due to large white margins. That should be improved in the final manuscript.

**AC12:** Done.

**RC13:** Tables 4 and 5: The captions are overlapping with the table content. In general the table captions should be above the table.

**AC13:** Thanks for pointing this out. I have now moved all the Table caption above the tables.